# THE LOSS SURFACE AND EXPRESSIVITY OF DEEP CONVOLUTIONAL NEURAL NETWORKS

## ABSTRACT

We analyze the expressiveness and loss surface of practical deep convolutional neural networks (CNNs) with shared weights and max pooling layers. We show that such CNNs produce linearly independent features at a "wide" layer which has more neurons than the number of training samples. This condition holds e.g. for the VGG network. Furthermore, we provide for such wide CNNs necessary and sufficient conditions for global minima with zero training error. For the case where the wide layer is followed by a fully connected layer we show that almost every critical point of the empirical loss is a global minimum with zero training error. Our analysis suggests that both depth and width are very important in deep learning. While depth brings more representational power and allows the network to learn high level features, width smoothes the optimization landscape of the loss function in the sense that a sufficiently wide network has a well-behaved loss surface with almost no bad local minima.

## 1 INTRODUCTION

It is well known that the optimization problem for training neural networks can have exponentially many local minima (Auer et al., 1996; Safran & Shamir, 2016) and NP-hardness has been shown in many cases (Blum & Rivest., 1989; Sima, 2002; Livni et al., 2014; Shamir, 2017; Shalev-Shwartz et al., 2017). However, it has been empirically observed (Dauphin et al., 2014; Goodfellow et al., 2015) that the training of state-of-the-art deep CNNs (LeCun et al., 1990; Krizhevsky et al., 2012), which are often overparameterized, is not hampered by suboptimal local minima.

In order to explain the apparent gap between hardness results and practical performance, many interesting theoretical results have been recently developed (Andoni et al., 2014; Sedghi & Anandkumar, 2015; Janzamin et al., 2016; Haeffele & Vidal, 2015; Gautier et al., 2016; Brutzkus & Globerson, 2017; Soltanolkotabi, 2017; Soudry & Hoffer, 2017; Goel & Klivans, 2017; Du et al., 2017; Zhong et al., 2017; Tian, 2017; Li & Yuan, 2017) in order to identify conditions under which one can guarantee that local search algorithms like gradient descent converge to the globally optimal solution. However, it turns out that these approaches are either not practical as they require e.g. knowledge about the data generating measure, or a modification of network structure and objective, or they are for quite restricted network structures, mostly one hidden layer networks, and thus are not able to explain the success of deep networks in general. For deep linear networks one has achieved a quite complete picture of the loss surface as it has been shown that every local minimum is a global minimum (Baldi & Hornik, 1988; Kawaguchi, 2016; Freeman & Bruna, 2017; Hardt & Ma, 2017; Yun et al., 2017). By randomizing the nonlinear part of a feedforward network with ReLU activation function and making some additional simplifying assumptions, Choromanska et al. (2015a) can relate the loss surface of neural networks to a certain spin glass model. In this model the objective of local minima is close to the global optimum and the number of bad local minima decreases quickly with the distance to the global optimum. This is a very interesting result but is based on a number of unrealistic assumptions (Choromanska et al., 2015b). More recently, Nguyen & Hein (2017) have analyzed deep fully connected networks with general activation functions and could show that almost every critical point is a global minimum if one layer has more neurons than the number of training points. While this result holds for networks in practice, it requires a quite extensively overparameterized network.

In this paper we overcome the restriction of previous work in several ways. This paper is one of the first ones, which theoretically analyzes deep CNNs. CNNs are of high practical interest as they learn very useful representations (Zeiler & Fergus, 2014; Mahendran & Vedaldi, 2015; Yosinski et al., 2015) with a small number of parameters. We are only aware of Cohen & Shashua (2016) who study the expressiveness of CNNs with max-pooling layer and ReLU activation but with rather unrealistic filters (just $1 \times 1$) and no shared weights. In our setting we allow as well max pooling and general activation functions. Moreover, we can have an arbitrary number of filters and we study general convolutions as the filters need not be applied to regular structures like $3 \times 3$ but can be patch-based where the only condition is that all the patches have the size of the filter. Convolutional layers, fully connected layers and max-pooling layers can be combined in almost arbitrary order. We study in this paper the expressiveness and loss surface of such a CNN where one layer is wide, in the sense that it has more neurons than the number of training points. While this assumption sounds at first quite strong, we want to emphasize that the VGG network (Simonyan & Zisserman, 2015) and other CNNs, see Table 3, fulfill this condition. We show that such wide CNNs produce linearly independent feature representations at the wide layer and thus are able to fit the training data exactly (universal finite sample expressivity). This is even true with probability one when all the parameters of the bottom layers (from input to the wide layer) are chosen randomly [1]. We think that this explains partially the results of Zhang et al. (2017) where they show experimentally for several CNNs that they are able to fit random labels. Moreover, we provide necessary and sufficient conditions for global minima with zero squared loss and show for a particular class of CNNs that almost all critical points are globally optimal, which to some extent explains why such wide CNNs can be optimized so efficiently.

## 2 DEEP CONVOLUTIONAL NEURAL NETWORKS

We first introduce our notation and definition of CNNs. Let $N$ be the number of training samples and denote by $X = [x_1, \ldots, x_N]^T \in \mathbb{R}^{N \times d}, Y = [y_1, \ldots, y_N]^T \in \mathbb{R}^{N \times m}$ the input resp. output matrix for the training data $(x_i, y_i)_{i=1}^N$, where $d$ is the input dimension and $m$ the number of classes.

Let $L$ be the number of layers of the network, where each layer is either a convolutional, max-pooling or fully connected layer. The layers are indexed from $k = 0, 1, \ldots, L$ which corresponds to input layer, 1st hidden layer, …, and output layer. Let $n_k$ be the width of layer $k$ and $f_k : \mathbb{R}^d \to \mathbb{R}^{n_k}$ the function that computes for every input its feature vector at layer $k$.

The convolutional layer consists of a set of patches of equal length where every patch is a subset of neurons from the same layer. Throughout this paper, we assume that the patches of every layer cover the whole layer, *i.e.* every neuron belongs to at least one of the patches, and that there are no patches that contain exactly the same subset of neurons. This means that if one patch covers the whole layer then it must be the only patch of the layer. Let $P_k$ and $l_k$ be the number of patches resp. the size of each patch at layer $k$ for every $0 \leq k < L$. For every input $x \in \mathbb{R}^d$, let $\{x^1, \ldots, x^{P_0}\} \in \mathbb{R}^{l_0}$ denote the set of patches at the input layer and $\left\{f_k^1(x), \ldots, f_k^{P_k}(x)\right\} \in \mathbb{R}^{l_k}$ the set of patches at layer $k$. Each filter of the layer consists of the same set of patches. We denote by $T_k$ the number of convolutional filters and by $W_k = [w_k^1, \ldots, w_k^{T_k}] \in \mathbb{R}^{l_{k-1} \times T_k}$ the corresponding parameter matrix of the convolutional layer $k$ for every $1 \leq k < L$. Each column of $W_k$ corresponds to one filter. Furthermore, $b_k \in \mathbb{R}^{n_k}$ denotes the bias vector and $\sigma_k : \mathbb{R} \to \mathbb{R}$ the activation function for each layer. Note that one can use the same activation function for all layers but we use the general form to highlight the role of different layers. In this paper, all functions are applied componentwise, and we denote by $[a]$ the set of integers $\{1, 2, \ldots, a\}$ and by $[a, b]$ the set of integers from $a$ to $b$.

**Definition 2.1 (Convolutional layer)** *A layer $k$ is called a convolutional layer if its output $f_k(x) \in \mathbb{R}^{n_k}$ is defined for every $x \in \mathbb{R}^d$ as*

$$f_k(x)_h = \sigma_k\Big( \big\langle w_k^t, f_{k-1}^p(x) \big\rangle + (b_k)_h \Big) \tag{1}$$

*for every $p \in [P_{k-1}], t \in [T_k], h := (p-1)T_k + t$.*

---

[1] for any probability measure on the parameter space which has a density with respect to the Lebesgue measure

The value of each neuron at layer $k$ is computed by first taking the inner product between a filter of layer $k$ and a patch at layer $k - 1$, adding the bias and then applying the activation function. The number of neurons at layer $k$ is thus $n_k = T_k P_{k-1}$, which we denote as the width of layer $k$. Our definition of a convolutional layer is quite general as every patch can be an arbitrary subset of neurons of the same layer and thus covers most of existing variants in practice.

Definition 2.1 includes the fully connected layer as a special case by using $P_{k-1} = 1, l_{k-1} = n_{k-1}, f_{k-1}^1(x) = f_{k-1}(x) \in \mathbb{R}^{n_{k-1}}, T_k = n_k, W_k \in \mathbb{R}^{n_{k-1} \times n_k}, b_k \in \mathbb{R}^{n_k}$. Thus we have only one patch which is the whole feature vector at this layer.

**Definition 2.2 (Fully connected layer)** *A layer $k$ is called a fully connected layer if its output $f_k(x) \in \mathbb{R}^{n_k}$ is defined for every $x \in \mathbb{R}^d$ as*

$$f_k(x) = \sigma_k \Big( W_k^T f_{k-1}(x) + b_k \Big). \tag{2}$$

For some results we also allow max-pooling layers.

**Definition 2.3 (Max-pooling layer)** *A layer $k$ is called a max-pooling layer if its output $f_k(x) \in \mathbb{R}^{n_k}$ is defined for every $x \in \mathbb{R}^d$ and $p \in [P_{k-1}]$ as*

$$f_k(x)_p = \max \Big( (f_{k-1}^p(x))_1, \ldots, (f_{k-1}^p(x))_{l_{k-1}} \Big). \tag{3}$$

A max-pooling layer just computes the maximum element of every patch from the previous layer. Since there are $P_{k-1}$ patches at layer $k - 1$, the number of output neurons at layer $k$ is $n_k = P_{k-1}$.

**Reformulation of Convolutional Layers:** For each convolutional or fully connected layer, we denote by $\mathcal{M}_k : \mathbb{R}^{l_{k-1} \times T_k} \to \mathbb{R}^{n_{k-1} \times n_k}$ the linear map that returns for every parameter matrix $W_k \in \mathbb{R}^{l_{k-1} \times T_k}$ the corresponding full weight matrix $U_k = \mathcal{M}_k(W_k) \in \mathbb{R}^{n_{k-1} \times n_k}$. For convolutional layers, $U_k$ can be seen as the counterpart of the weight matrix $W_k$ in fully connected layers. We define $U_k = \mathcal{M}_k(W_k) = W_k$ if layer $k$ is fully connected. Note that the mapping $\mathcal{M}_k$ depends on the patch structure of each convolutional layer $k$. For example, suppose that layer $k$ has two filters of length 3, that is, $W_k = [w_k^1, w_k^2] = \begin{bmatrix} a & d \\ b & e \\ c & f \end{bmatrix}$, and $n_{k-1} = 5$ and patches given by a 1D-convolution with stride 1 and no padding then: $U_k^T = \mathcal{M}_k(W_k)^T = \begin{bmatrix} a & b & c & 0 & 0 \\ d & e & f & 0 & 0 \\ 0 & a & b & c & 0 \\ 0 & d & e & f & 0 \\ 0 & 0 & a & b & c \\ 0 & 0 & d & e & f \end{bmatrix}$,

The above ordering of the rows of $U_k^T$ of a convolutional layer is determined by Equation (1), in particular, the row index $h$ of $U_k^T$ is calculated as $h = (p-1)T_k + t$, which means for every given patch $p$ one has to loop over all the filters $t$ and compute the corresponding value of the output unit by taking the inner product of the $h$-th row of $U_k^T$ with the whole feature vector of the previous layer.

We assume throughout this paper that that there is no non-linearity at the output layer. By ignoring max-pooling layers for the moment, the feature maps $f_k : \mathbb{R}^d \to \mathbb{R}^{n_k}$ can be written as

$$f_0(x) = x, \qquad f_k(x) = \sigma_k \big( g_k(x) \big), \text{ where } g_k(x) = U_k^T f_{k-1}(x) + b_k, \quad \forall 1 \le k \le L - 1$$

$$f_L(x) = g_L(x) = U_L^T f_{L-1}(x) + b_L,$$

where $g_k : \mathbb{R}^d \to \mathbb{R}^{n_k}$ is the pre-activation output at layer $k$. By stacking the feature vectors of layer $k$ of all training samples, before and after applying the activation function, into a matrix, we define:

$$F_k = [f_k(x_1), \ldots, f_k(x_N)]^T \in \mathbb{R}^{N \times n_k}, \quad \text{and} \quad G_k = [g_k(x_1), \ldots, g_k(x_N)]^T \in \mathbb{R}^{N \times n_k}.$$

In this paper, we refer to $F_k$ as the output matrix at layer $k$. It follows from above that

$$F_0 = X, \qquad F_k = \sigma_k(G_k), \text{ where } G_k = F_{k-1} U_k + \mathbf{1}_N b_k^T, \quad \forall 1 \le k \le L - 1 \tag{4}$$

$$F_L = G_L = F_{L-1} U_L + \mathbf{1}_N b_L^T. \tag{5}$$

In this paper, we assume the following general condition on the structure of convolutional layers.

**Assumption 2.4 (Convolutional Structure)** *For every convolutional layer $k$, there exists at least one parameter matrix $W_k \in \mathbb{R}^{l_{k-1} \times T_k}$ for which the corresponding weight matrix $U_k = \mathcal{M}_k(W_k) \in \mathbb{R}^{n_{k-1} \times n_k}$ has full rank.*

It is straightforward to see that Assumption 2.4 is satisfied if every neuron of a convolutional layer belongs to at least one patch and there are no identical patches.

**Lemma 2.5** *If Assumption 2.4 holds, then for every convolutional layer $k$, the set of $W_k \in \mathbb{R}^{l_{k-1} \times T_k}$ for which $U_k = \mathcal{M}_k(W_k) \in \mathbb{R}^{n_{k-1} \times n_k}$ does not have full rank has Lebesgue measure zero.*

**Proof:** Since $U_k = \mathcal{M}_k(W_k) \in \mathbb{R}^{n_{k-1} \times n_k}$ and $\mathcal{M}_k$ is a linear map, every entry of $U_k$ is a linear function of the entries of $W_k$. Let $m = \min(n_{k-1}, n_k)$, then the set of low rank matrices $U_k$ is characterized by a system of equations where the $\binom{\max(n_{k-1}, n_k)}{m}$ determinants of all $m \times m$ sub-matrices of $U_k$ are zero. As the determinant is a polynomial in the entries of the matrix and thus a real analytic function, and the composition of analytic functions is again analytic, we get that each determinant is a real analytic function of $W_k$. By Assumption 2.4, there exists at least one $W_k$ such that one of these determinants is non-zero. Thus by Lemma A.2, the set of $W_k$ where this determinant is zero has Lebesgue measure zero. As all the submatrices need to have low rank in order that $U_k$ has low rank, we get that the set of $W_k$ where $U_k$ has low rank has Lebesgue measure zero. $\square$

## 3    WIDE CNNS CAN LEARN LINEARLY INDEPENDENT FEATURES

In this section, we show that a class of standard CNN architectures with convolutional layers, fully connected layers and max-pooling layers plus standard activation functions like ReLU, sigmoid, softplus, etc are able to learn linearly independent features at any hidden layer if that layer has more neurons than the number of training samples. Our assumption on training data is the following.

**Assumption 3.1 (Training data)** *The patches of different training samples are non-identical, that is, $x_i^p \neq x_j^q$ for every $p, q \in [P_0], i, j \in [N], i \neq j$.*

Assumption 3.1 is quite weak, especially if the size of the input patches is larger. If the assumption does not hold, one can add a small perturbation to the training samples: $\{x_1 + \epsilon_1, \ldots, x_N + \epsilon_N\}$. The set of $\{\epsilon_i\}_{i=1}^N$ where Assumption 3.1 is not fulfilled for the new dataset has measure zero. Moreover, $\{\epsilon_i\}_{i=1}^N$ can be chosen arbitrarily small so that the influence of the perturbation is negligible. Our main assumptions on the activation function of the hidden layers is the following.

**Assumption 3.2 (Activation function)** *The activation function $\sigma$ is continuous, non-constant, and satisfies one of the following conditions:*

- *$\exists \mu_+, \mu_- \in \mathbb{R}$ s.t. $\lim_{t \to -\infty} \sigma_k(t) = \mu_-$ and $\lim_{t \to \infty} \sigma_k(t) = \mu_+$ and $\mu_+ \mu_- = 0$*

- *$\exists \rho_1, \rho_2, \rho_3, \rho_4 \in \mathbb{R}_+$ s.t. $|\sigma(t)| \leq \rho_1 e^{\rho_2 t}$ for $t < 0$ and $|\sigma(t)| \leq \rho_3 t + \rho_4$ for $t \geq 0$*

Assumption 3.2 covers several standard activation functions.

**Lemma 3.3** *The following activation functions satisfy Assumption 3.2:*

*ReLU: $\sigma(t) = \max(0, t)$, Sigmoid: $\sigma(t) = \dfrac{1}{1 + e^{-t}}$, Softplus: $\sigma_\alpha(t) = \dfrac{1}{\alpha} \ln(1 + e^{\alpha t})$ for $\alpha > 0$.*

The softplus function is a smooth approximation of ReLU. It holds:

$$\lim_{\alpha \to \infty} \sigma_\alpha(t) = \lim_{\alpha \to \infty} \frac{1}{\alpha} \ln(1 + e^{\alpha t}) = \max(0, t). \tag{6}$$

The first main result of this paper is the following.

**Theorem 3.4 (Linearly Independent Features)** *Let Assumption 3.1 hold for the training sample. Consider a deep CNN architecture for which there exists some layer $1 \leq k \leq L - 1$ such that*

1. *Layer* 1 *and layer* $k$ *are convolutional or fully connected while all the other layers can be convolutional, fully connected or max-pooling*

2. *The width of layer* $k$ *is larger than the number of training samples,* $n_k = T_k P_{k-1} \geq N$

3. $(\sigma_1, \ldots, \sigma_k)$ *satisfy Assumption 3.2*

*Then there exist a set of parameters of the first* $k$ *layers* $(W_l, b_l)_{l=1}^k$ *such that the set of feature vectors* $\{f_k(x_1), \ldots, f_k(x_N)\}$ *are linearly independent. Moreover,* $(W_l, b_l)_{l=1}^k$ *can be chosen in such a way that all the weight matrices* $U_l = \mathcal{M}_l(W_l)$ *have full rank for every* $1 \leq l \leq k.$

Theorem 3.4 implies that a large class of CNNs employed in practice with convolutional, fully connected and max-pooling layers and standard activation functions like ReLU, sigmoid or softplus can produce linearly independent features at any hidden layer if its width is larger than the number of training samples. Figure 1 shows an example of a CNN architecture that satisfies the conditions of Theorem 3.4 at the first convolutional layer (*i.e.* $k = 1$). Note that if a set of vectors is linearly independent then they are also linearly separable. In this sense, Theorem 3.4 suggests that deep and wide CNNs can produce linearly separable features at every wide hidden layer.

Linear separability in neural networks has been recently studied by An et al. (2015), where the authors show that a two-hidden-layer fully connected network with ReLU activation function can transform any training set to be linearly separable while approximately preserving the distances of the training data at the output layer. Compared to An et al. (2015) our Theorem 3.4 is derived for CNNs with a wider range of activation functions. Moreover, our result shows even linear independence of features which is stronger than linear separability. Recently, Nguyen & Hein (2017) have shown a similar result for deep fully connected networks and analytic activation functions.

We note that, in contrast to fully connected networks, for CNNs the condition $n_k \geq N$ of Theorem 3.4 does not necessarily imply that the network has a huge number of parameters as the layers $k$ and $k + 1$ can be chosen to be convolutional. In particular, the condition $n_k = T_k P_{k-1} \geq N$ can be fulfilled by increasing the number of filters $T_k$ or by using a large number of patches $P_{k-1}$ (however $P_{k-1}$ is upper bounded by $n_k$), which is however only possible if $l_{k-1}$ is small as otherwise our condition on the patches cannot be fulfilled. Interestingly, such a architecture has been used in the VGG-Net (Simonyan & Zisserman, 2015), where they use small $3 \times 3$ filters and minimal stride 1 in the first layer and thus they fulfill the condition $n_k \geq N$ for $k = 1$, see Table 3, for ImageNet. Also note that other state-of-the-art-networks fulfill the condition in Table 3. Overall, Theorem 3.4 can be seen as a theoretical support for the usage of small filters and strides in practical CNN architectures as it increases the chance of achieving linear separability at early hidden layers in the network and also reduces the total number of training parameters. The reason why linear separability helps will be discussed in Section 4 when we analyze the loss surface of the CNNs. Note also that the condition $n_k \geq N$ is a sufficient condition but not necessary to prove our results. In particular, we conjecture that linear separability might hold with far less number of neurons in practical applications.

One might ask now how difficult it is to find such parameters which generate linearly independent features at a hidden layer? Our next result shows that once analytic[2] activation functions, *e.g.* sigmoid or softplus, are used at the first $k$ hidden layers of the network, the linear independence of features at layer $k$ holds with probability 1 even if one draws the parameters of the first $k$ layers $(W_l, b_l)^k$ randomly for any probability measure on the parameter space which has a density with respect to the Lebesgue measure.

**Theorem 3.5** *Let Assumption 3.1 hold for the training samples. Consider a deep CNN for which there exists some layer* $1 \leq k \leq L - 1$ *such that*

1. *Every layer from* 1 *to* $k$ *is convolutional or fully connected*

2. *The width of layer* $k$ *is larger than number of training samples, that is,* $n_k = T_k P_{k-1} \geq N$

3. $(\sigma_1, \ldots, \sigma_k)$ *are real analytic functions and satisfy Assumption 3.2.*

---

[2]A function $\sigma : \mathbb{R} \to \mathbb{R}$ is real analytic if its Taylor series about $x_0$ converges to $\sigma(x_0)$ on some neighborhood of $x_0$ for every $x_0 \in \mathbb{R}$ (Krantz & Parks, 2002).

Table 1: The smallest singular value $\sigma_{\min}(F_1)$ of the feature matrix $F_1$ of the first convolutional layer (similar $\sigma_{\min}(F_3)$ for the feature matrix $F_3$ of the second convolutional layer) of the trained network in Figure 1 are shown for varying number of convolutional filters $T_1$. The rank of a matrix $A \in \mathbb{R}^{m \times n}$ is estimated (see Chapter 2.6.1 in Press (2007)) by computing the singular value decomposition of $A$ and counting the singular values which exceed the threshold $\frac{1}{2}\sqrt{m+n+1}\,\sigma_{\max}(A)\epsilon$, where $\epsilon$ is machine precision. For all filter sizes the feature matrices $F_1$ have full rank.

| $T_1$ | size($F_1$) | rank($F_1$) | $\sigma_{\min}(F_1)$ | size($F_3$) | rank($F_3$) | $\sigma_{\min}(F_3)$ |
|---|---|---|---|---|---|---|
| 10 | $60000 \times 6760$ | **6760** | $3.7 \times 10^{-6}$ | $60000 \times 2880$ | **2880** | $2.0 \times 10^{-2}$ |
| 20 | $60000 \times 13520$ | **13520** | $2.2 \times 10^{-6}$ | $60000 \times 2880$ | **2880** | $7.0 \times 10^{-4}$ |
| 30 | $60000 \times 20280$ | **20280** | $1.5 \times 10^{-6}$ | $60000 \times 2880$ | **2880** | $2.4 \times 10^{-4}$ |
| 40 | $60000 \times 27040$ | **27040** | $2.0 \times 10^{-6}$ | $60000 \times 2880$ | **2880** | $2.2 \times 10^{-3}$ |
| 50 | $60000 \times 33800$ | **33800** | $1.3 \times 10^{-6}$ | $60000 \times 2880$ | **2880** | $3.9 \times 10^{-5}$ |
| 60 | $60000 \times 40560$ | **40560** | $1.1 \times 10^{-6}$ | $60000 \times 2880$ | **2880** | $4.0 \times 10^{-5}$ |
| 70 | $60000 \times 47320$ | **47320** | $7.5 \times 10^{-7}$ | $60000 \times 2880$ | **2880** | $7.1 \times 10^{-3}$ |
| 80 | $60000 \times 54080$ | **54080** | $5.4 \times 10^{-7}$ | $60000 \times 2880$ | 2875 | $4.9 \times 10^{-18}$ |
| 89 | $60000 \times 60164$ | **60000** | $2.0 \times 10^{-8}$ | $60000 \times 2880$ | **2880** | $8.9 \times 10^{-10}$ |
| 100 | $60000 \times 67600$ | **60000** | $1.1 \times 10^{-6}$ | $60000 \times 2880$ | 2856 | $8.5 \times 10^{-27}$ |

Table 2: The loss and the number of misclassified training and test samples (train/test errors) of the corresponding trained networks in Table 1. Zero training error is attained for $T_1 \geq 30$.

| $T_1$ | 10 | 20 | 30 | 40 | 50 | 60 | 70 | 80 | 89 | 100 |
|---|---|---|---|---|---|---|---|---|---|---|
| Train loss ($\times 10^{-5}$) | 2.4 | 1.2 | 0.24 | 0.62 | 0.02 | 0.57 | 0.12 | 0.11 | 0.35 | 0.04 |
| Train error | 8 | 1 | **0** | **0** | **0** | **0** | **0** | **0** | **0** | **0** |
| Test error | 151 | 132 | 174 | 124 | 143 | 141 | 120 | 140 | 117 | 139 |

*Then the set of parameters of the first $k$ layers $(W_l, b_l)_{l=1}^k$ for which the set of feature vectors $\{f_k(x_1), \ldots, f_k(x_N)\}$ of layer $k$ are **not** linearly independent has Lebesgue measure zero.*

Note that Theorem 3.5 is a much stronger statement than Theorem 3.4, as it shows that for almost all weight configurations one gets linearly independent features at the wide layer. While Theorem 3.5 does not hold for the ReLU activation function as it is not an analytic function, we want to note again that one can approximate the ReLU function arbitrarily well using the softplus function (see Equation 6), which is an analytic function for any $\alpha > 0$ and thus Theorem 3.5 applies. It is an open question if the result holds also for the ReLU activation function itself.

To illustrate Theorem 3.5 we plot the rank of the feature matrices of the network in Figure 1. We use the standard benchmark MNIST dataset with $N = 60000$ training samples and 10000 test samples. First, we add small Gaussian noise $\mathcal{N}(0, 10^{-5})$ to every training sample so that Assumption 3.1 is fulfilled. We then vary the number of convolutional filters $T_1$ of the first layer from 10 to 100 and train the corresponding network with squared loss and sigmoid activation function using Adam (Kingma & Ba, 2015) with default hyperparameters and decaying learning rate for 2000 epochs. We present the smallest singular value of the feature matrices in Table 1 and the corresponding training loss, training error and test error in Table 2. If number of convolutional filters is large enough (*i.e.* $T_1 \geq 89$) it holds that $n_1 = 26 \times 26 \times T_1 \geq N = 60000$, and the second condition of Theorem 3.5 is satisfied for $k = 1$. Table 1 shows that the feature matrices $F_1$ have full rank in all cases (and $F_3$ in almost all cases), in particular for $T_1 \geq 89$ as shown in Theorem 3.5. As expected when the training samples are linearly independent after the first layer ($F_1$ has rank 60000 for $T \geq 89$) the training error is zero and the training loss is close to zero (note that the GPU uses single precision). However, note that linear independence is much stronger than linear separability and thus even for $T < 89$ one can achieve already zero training error and loss and thus much smaller number of neurons suffice.

It is very interesting to note that Theorem 3.5 explains previous empirical observations. In particular, Czarnecki et al. (2017) have shown empirically that linear separability is often obtained already in the first few hidden layers of the trained networks. This is done by attaching a linear classifier probe

(Alain & Bengio, 2016) to every hidden layer in the network after training the whole network with backpropagation. The fact that Theorem 3.5 holds even if the parameters of the bottom layers up to the wide layer $k$ are chosen randomly is also in line with recent empirical observations for CNN architectures that one has little loss in performance if the weights of the initial layers are chosen randomly without training (Jarrett et al., 2009; Saxe et al., 2011; Yosinski et al., 2014).

An application of Theorem 3.4 yields the following universal finite sample expressivity for CNNs. In particular, a deep CNN architecture with scalar output can perfectly express the values of any scalar-valued function over a finite number of inputs as long as the width of the last hidden layer is larger than the number of training samples.

**Corollary 3.6 (Universal Finite Sample Expressivity)** *Let Assumption 3.1 hold for the training samples. Consider a standard CNN with scalar output which satisfies the conditions of Theorem 3.4 at the last hidden layer $k = L - 1$. Let $f_L : \mathbb{R}^d \to \mathbb{R}$ be the output of the network given as*

$$f_L(x) = \sum_{j=1}^m \lambda_j f_{(L-1)j}(x) \quad \forall x \in \mathbb{R}^d$$

*where $\lambda \in \mathbb{R}^{n_{L-1}}$ is the weight vector of the last layer. Then for every target output $y \in \mathbb{R}^N$, there exists $\left\{ \lambda, (W_l, b_l)_{l=1}^{L-1} \right\}$ so that it holds $f_L(x_i) = y_i$ for every $i \in [N]$.*

**Proof:** Since the network satisfies the conditions of Theorem 3.4 for $k = L - 1$, there exists a set of parameters $(W_l, b_l)_{l=1}^{L-1}$ such that $rank(F_{L-1}) = N$. Let $F_L = [f_L(x_1), \dots, f_L(x_N)]^T \in \mathbb{R}^N$ then it follows that $F_L = F_{L-1}\lambda$. Pick $\lambda = F_{L-1}^T(F_{L-1}F_{L-1}^T)^{-1}y$ then it holds $F_L = F_{L-1}\lambda = y$. □

The expressivity of neural networks has been well-studied in the literature, in particular in the universal approximation theorems for one hidden layer networks (Cybenko, 1989; Hornik et al., 1989). Recently, many results have be shown why deep networks are superior to shallow networks in terms of expressiveness (Delalleau & Bengio, 2011; Telgarsky, 2016; 2015; Eldan & Shamir, 2016; Safran & Shamir, 2017; Yarotsky, 2016; Poggio et al., 2016; Liang & Srikant, 2017; Mhaskar & Poggio, 2016; Montufar et al., 2014; Pascanu et al., 2014; Raghu et al., 2017). While most of these results are derived for fully connected networks, it seems that Cohen & Shashua (2016) are the first ones who study expressivity of CNNs. In particular, they show that CNNs with max-pooling and ReLU units are universal in the sense that they can approximate any given function if the size of the networks is unlimited. However, the number of convolutional filters in this result has to grow exponentially with the number of patches and they do not allow shared weights in their result, which is a standard feature of CNNs. Our Corollary 3.6 shows universal finite sample expressivity, instead of universal function approximation, even for $L = 2$ and $k = 1$, that is a single convolutional layer network can perfectly fit the training data as long as the number of hidden units is not smaller than the number of samples. To the best of our knowledge, this is the first result on universal finite sample expressivity for a large class of practical CNNs.

For fully connected networks, universal finite sample expressivity has been studied by Zhang et al. (2017); Nguyen & Hein (2017); Hardt & Ma (2017). They show that a network with a single hidden layer with $N$ hidden units can express any training set of size $N$. While the number of training parameters of a single hidden layer CNN with $N$ hidden units and scalar output is just $2N + T_1 l_0$, where $T_1$ is the number of convolutional filters and $l_0$ is the length of each filter, it is $Nd + 2N$ for fully connected networks. If we set the width of the hidden layer of the CNN as $n_1 = T_1 P_0 = N$ in order to fulfill the condition of Corollary 3.6, then the number of training parameters of the CNN becomes $2N + N l_0 / P_0$, which is less than $3N$ if $l_0 \leq P_0$ compared to $(d + 2)N$ for the fully connected case. In practice one almost always has $l_0 \leq P_0$ as $l_0$ is typically a small integer and $P_0$ is on the order of the dimension of the input. Thus, the number of parameters of the CNN to achieve universal finite sample expressivity is significantly smaller than that of fully connected networks.

Obviously, in practice it is more important that the network generalizes rather than just fitting the training data. By using shared weights and sparsity structure, CNNs seem to implicitly regularize the model class in order to achieve good generalization performance. Thus even though they can fit also random labels or noise (Zhang et al., 2017) due to the universal finite sample expressivity shown in Corollary 3.6, they seem still to be able to generalize well (Zhang et al., 2017).

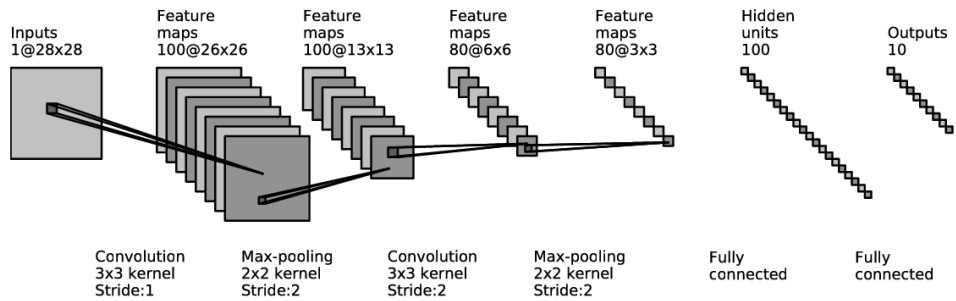

Figure 1: An example of CNN for a given training set of size $N \leq 100 \times 26 \times 26 = 67600$. The width of each layer is $d = n_0 = 784, n_1 = 67600, n_2 = 16900, n_3 = 2880, n_4 = 720, n_5 = 100, n_6 = m = 10$. One can see that $n_1 \geq N$ and the network has pyramidal structure from layer 2 till the output layer, that is, $n_2 \geq \ldots \geq n_6$.

Table 3: The width of the first convolutional layer ($n_1$) and the maximum width of all the hidden layers ($\max_{1 \leq k \leq L-1} n_k$) of state-of-the-art CNN architectures in comparison with the size of ImageNet ($N \approx 1200K$). All numbers are lower bounds on the true width.

| CNN Architecture | $n_1$ | $\max_k n_k$ |
|---|---|---|
| VGG(A-E) (Simonyan & Zisserman, 2015) | 3000K | 3000K |
| InceptionV3 (Szegedy et al., 2015b) | 700K | 1300K |
| InceptionV4 (Szegedy et al., 2016) | 700K | 1300K |
| SqueezeNet (Iandola et al., 2016) | 1180K | 1180K |
| Enet (Paszke et al., 2016) | 1000K | 1000K |
| GoogLeNet (Szegedy et al., 2015a) | 800K | 800K |
| ResNet (He et al., 2016) | 800K | 800K |
| Xception (Chollet, 2016) | 700K | 700K |

## 4 THE LOSS SURFACE OF CONVOLUTIONAL NEURAL NETWORKS

In this section, we restrict our analysis to the use of least squares loss. However, as we show later that the network can produce exactly the target output (*i.e.* $F_L = Y$) for some choice of parameters, all our results can also be extended to any other loss function where the global minimum is attained at $F_L = Y$, for instance the squared Hinge-loss analyzed in Nguyen & Hein (2017). Let $\mathcal{P}$ denote the space of all parameters of the network. The final training objective $\Phi : \mathcal{P} \to \mathbb{R}$ is given as

$$\Phi\left((W_l, b_l)_{l=1}^L\right) = \frac{1}{2} \|F_L - Y\|_F^2 \tag{7}$$

where $F_L$ is defined as in (5), which is also the same as

$$F_L = \sigma_{L-1}(\ldots \sigma_1(XU_1 + \mathbf{1}_N b_1^T) \ldots)U_L + \mathbf{1}_N b_L^T,$$

where $U_l = \mathcal{M}_l(W_l)$ for every $1 \leq l \leq L$.

Our assumptions on the architecture of CNNs is given below.

**Assumption 4.1 (CNN Architecture)** *Every layer in the network is a convolutional layer or fully connected layer and the output layer is fully connected. Moreover, there exists some hidden layer $1 \leq k \leq L-1$ such that the following holds:*

- *The width of layer $k$ is larger than number of training samples, that is, $n_k = T_k P_{k-1} \geq N$*

- *All the activation functions of the hidden layers $(\sigma_1, \ldots, \sigma_{L-1})$ satisfy Assumption 3.2*

- *$(\sigma_{k+1}, \ldots, \sigma_{L-1})$ are strictly increasing or strictly decreasing, and differentiable*

- *The network is pyramidal from layer $k+1$ till the output layer, that is, $n_{k+1} \geq \ldots \geq n_L$*

A typical example that satisfies Assumption 4.1 is the following (see Figure 1 for an illustration):

- The first layer is a convolutional layer with $n_1 = T_1 P_0 \geq N$
- Every layer from layer 2 till the output layer is convolutional or fully connected
- $(\sigma_1, \ldots, \sigma_k)$ can be ReLU, sigmoid or softplus
- $(\sigma_{k+1}, \ldots, \sigma_{L-1})$ can be sigmoid or softplus
- $n_2 \geq n_3 \geq \ldots \geq n_L$

One can easily check that the above example satisfies Assumption 4.1 for $k = 1$.

In the following, let us define for every $1 \leq k \leq L - 1$ the subset $S_k \subseteq \mathcal{P}$ such that

$$S_k := \left\{ (W_l, b_l)_{l=1}^{L} \mid rank(F_k) = N \text{ and } U_l \text{ has full rank for every } l \in [k+2, L] \right\}.$$

The set $S_k$ is the set of parameters where the feature vectors at layer $k$ are linearly independent and all the weight matrices from layer $k + 2$ till the output layer have full rank. In the following, we examine conditions for global optimality in $S_k$. It is important to note that $S_k$ covers almost the whole parameter space under an additional mild condition on the activation function.

**Lemma 4.2** *Let Assumption 3.1 hold for the training sample and let the CNN architecture satisfy Assumption 4.1 for some layer $1 \leq k \leq L - 1$. We assume further that the activation functions of the first $k$ layers $(\sigma_1, \ldots, \sigma_k)$ are real analytic. Then the set $\mathcal{P} \setminus S_k$ has Lebesgue measure zero.*

**Proof:** One can see that

$$\mathcal{P} \setminus S_k \subseteq \left\{ (W_l, b_l)_{l=1}^{L} \mid rank(F_k) < N \right\} \cup \left\{ (W_l, b_l)_{l=1}^{L} \mid U_l \text{ has low rank for some layer } l \right\}.$$

By Theorem 3.5, it holds that the set $\left\{ (W_l, b_l)_{l=1}^{L} \mid rank(F_k) < N \right\}$ has Lebesgue measure zero. Moreover, it follows from Lemma 2.5 that the set $\left\{ (W_l, b_l)_{l=1}^{L} \mid U_l \text{ has low rank for some layer } l \right\}$ also has measure zero. Thus, $\mathcal{P} \setminus S_k$ has Lebesgue measure zero. $\square$

In the next key lemma, we bound the objective function in terms of its gradient magnitude w.r.t. the weight matrix of layer $k$ for which $n_k \geq N$. For every matrix $A \in \mathbb{R}^{m \times n}$, let $\sigma_{\min}(A)$ and $\sigma_{\max}(A)$ denote the smallest and largest singular value of $A$. Let $\|A\|_F = \sqrt{\sum_{i,j} A_{ij}^2}$, $\|A\|_{\min} := \min_{i,j} |A_{ij}|$ and $\|A\|_{\max} := \max_{i,j} |A_{ij}|$. From Equations (4), (5) and (7), it follows that $\Phi$ can be seen as a function of $(U_l, b_l)_{l=1}^{L}$, and thus we can use $\nabla_{U_k} \Phi$. If layer $k$ is fully connected then $U_k = \mathcal{M}_k(W_k) = W_k$ and thus $\nabla_{U_k} \Phi = \nabla_{W_k} \Phi$. Otherwise, if layer $k$ is convolutional then we note that $\nabla_{U_k} \Phi$ is "not" the true gradient of the training objective because $U_k$ is not the true optimization parameter but $W_k$. In this case, the true gradient of $\Phi$ w.r.t. to the true parameter matrix $W_k$ which consists of convolutional filters can be computed via chain rule as

$$\frac{\partial \Phi}{\partial (W_k)_{rs}} = \sum_{i,j} \frac{\partial \Phi}{\partial (U_k)_{ij}} \frac{\partial (U_k)_{ij}}{\partial (W_k)_{rs}}$$

Please note that even though we write the partial derivatives with respect to the matrix elements, $\nabla_{W_k} \Phi$ resp. $\nabla_{U_k} \Phi$ are the matrices of the same dimension as $W_k$ resp. $U_k$ in the following.

**Lemma 4.3** *Consider a standard deep CNN which satisfies Assumption 4.1 for some hidden layer $1 \leq k \leq L - 1$. Then it holds*

$$\left\| \nabla_{U_{k+1}} \Phi \right\|_F \geq \sigma_{min}(F_k) \left( \prod_{l=k+1}^{L-1} \sigma_{min}(U_{l+1}) \left\| \sigma_l'(G_l) \right\|_{min} \right) \left\| F_L - Y \right\|_F$$

*and*

$$\left\| \nabla_{U_{k+1}} \Phi \right\|_F \leq \sigma_{max}(F_k) \left( \prod_{l=k+1}^{L-1} \sigma_{max}(U_{l+1}) \left\| \sigma_l'(G_l) \right\|_{max} \right) \left\| F_L - Y \right\|_F.$$

Our next main result is motivated by the fact that empirically when training over-parameterized neural networks with shared weights and sparsity structure like CNNs, there seem to be no problems with sub-optimal local minima. In many cases, even when training labels are completely random, local search algorithms like stochastic gradient descent can converge to a solution with almost zero training error (Zhang et al., 2017). To understand better this phenomenon, we first characterize in the following Theorem 4.4 the set of points in parameter space with zero loss, and then analyze in Theorem 4.5 the loss surface for a special case of the network. We emphasize that our results hold for standard deep CNNs with convolutional layers with shared weights and fully connected layers.

**Theorem 4.4 (Necessary and Sufficient Condition for Zero Training Error)** *Let Assumption 3.1 hold for the training sample and suppose that the CNN architecture satisfies Assumption 4.1 for some hidden layer $1 \leq k \leq L - 1$. Let $\Phi : \mathcal{P} \to \mathbb{R}$ be defined as in (7). Given any point $(W_l, b_l)_{l=1}^L \in S_k$. Then it holds that $\Phi\left((W_l, b_l)_{l=1}^L\right) = 0$ if and only if $\nabla_{U_{k+1}} \Phi\Big|_{(W_l, b_l)_{l=1}^L} = 0$.*

**Proof:** If $\Phi\left((W_l, b_l)_{l=1}^L\right) = 0$ then it follows from the upper bound of Lemma 4.3 that $\nabla_{U_{k+1}} \Phi = 0$. Now, we suppose that $\nabla_{U_{k+1}} \Phi = 0$. Since $(W_l, b_l)_{l=1}^L \in S_k$ it holds $rank(F_k) = N$ and $U_l$ has full rank for every $l \in [k + 2, L]$. Thus it holds $\sigma_{\min}(F_k) > 0$ and $\sigma_{\min}(U_l) > 0$ for every $l \in [k + 2, L]$. Moreover, $(\sigma_{k+1}, \ldots, \sigma_{L-1})$ have non-zero derivative by Assumption 4.1 and thus $\|\sigma_l'(G_l)\|_{\min} > 0$ for every $l \in [k + 1, L - 1]$. This combined with the lower bound in Lemma 4.3 leads to $\|F_L - Y\|_F = 0$ and thus $\Phi\left(W_l, b_l)_{l=1}^L\right) = 0$. $\qquad \square$

Lemma 4.2 shows that the set of points which are not covered by Theorem 4.4 has just measure zero under a mild condition. The necessary and sufficient condition of Theorem 4.4 is rather intuitive as it requires the gradient of the training objective to vanish w.r.t. the full weight matrix of layer $k + 1$ regardless of the architecture of this layer. It turns out that if layer $k + 1$ is fully connected, then this condition is always satisfied at a critical point, in which case we obtain that every critical point in $S_k$ is a global minimum with exact zero training error. This is shown in the next Theorem 4.5, where we consider a classification task with $m$ classes. $Z \in \mathbb{R}^{m \times m}$ is the full rank class encoding matrix e.g. the identity matrix and $(X, Y)$ the training sample such that $Y_{i:} = Z_{j:}$ whenever $x_i$ belongs to class $j$ for every $i \in [N], j \in [m]$.

**Theorem 4.5 (Loss Surface of CNNs)** *Let $(X, Y, Z)$ be a training set for which Assumption 3.1 holds, the CNN architecture satisfies Assumption 4.1 for some hidden layer $1 \leq k \leq L - 1$, and layer $k + 1$ is fully connected. Let $\Phi : \mathcal{P} \to \mathbb{R}$ be defined as in (7). Then*

- *Every critical point $(W_l, b_l)_{l=1}^L \in S_k$ is a global minimum with $\Phi\left((W_l, b_l)_{l=1}^L\right) = 0$*

- *There exist infinitely many global minima $(W_l, b_l)_{l=1}^L \in S_k$ with $\Phi\left((W_l, b_l)_{l=1}^L\right) = 0$*

Theorem 4.5 indicates that the loss surface for this type of CNNs has a rather simple structure in the sense that almost every critical point is a global minimum with zero training error. It remains an interesting open problem if this result can be transferred to the case where layer $k + 1$ is also convolutional. In any case whether layer $k + 1$ is fully connected or not, one might still assume that a solution with zero training error still exists. However, note that Theorem 4.4 shows that at those points where the loss is zero, the gradient of $\Phi$ w.r.t. $U_{k+1}$ must be zero as well. An interesting special case of Theorem 4.5 is when the network is fully connected, in which case all the results of Theorem 4.5 hold without any modifications.

**Corollary 4.6 (Loss Surface of Fully Connected Nets)** *Let $(X, Y, Z)$ be a training set with non-identical training samples, i.e. $x_i \neq x_j$ for every $i \neq j$ and the fully connected network satisfies Assumption 4.1 for some layer $1 \leq k \leq L - 1$. Let $\Phi : \mathcal{P} \to \mathbb{R}$ be defined as in (7). Then the following holds*

- *Every critical point $(W_l, b_l)_{l=1}^L \in S_k$ is a global minimum with $\Phi\left((W_l, b_l)_{l=1}^L\right) = 0$*

- *There exist infinitely many global minima $(W_l, b_l)_{l=1}^L \in S_k$ with $\Phi\left((W_l, b_l)_{l=1}^L\right) = 0$*

Corollary 4.6 can be seen as a formal proof for the implicit assumption used in the recent work (Nguyen & Hein, 2017) that there exists a global minimum with zero training error for the class of fully connected, deep and wide networks.

## 5 CONCLUSION

We have analyzed the expressiveness and loss surface of CNNs in realistic and practically relevant settings. As state-of-the-art networks fulfill exactly or approximately the condition to have a sufficiently wide convolutional layer, we think that our results help to understand why current CNNs can be trained so effectively. It would be interesting to discuss the loss surface for cross-entropy loss, which currently does not fit into our analysis as the global minimum does not exist when the data is linearly separable.

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

## A    MISSING PROOFS OF SECTION 3

### A.1    PROOF OF LEMMA 3.3

- ReLU: It holds for every $t < 0$ that $\sigma(t) = \max(0, t) = 0 < e^t$, and for $t \geq 0$ that $\sigma(t) = t < t + 1$. Thus ReLU satisfies the second condition of Assumption 3.2.

- Sigmoid: It holds that

$$\lim_{t \to -\infty} \frac{1}{1 + e^{-t}} = 0, \quad \lim_{t \to \infty} \frac{1}{1 + e^{-t}} = 1.$$

  Thus $\sigma$ satisfies the first condition of Assumption 3.2.

- Softplus: Since $1 + e^{\alpha t} \leq 2e^{\alpha t}$ for every $t \geq 0$, it holds for every $t \geq 0$ that

$$0 \leq \sigma_\alpha(t) = \frac{1}{\alpha} \log(1 + e^{\alpha t}) \leq \frac{1}{\alpha} \log(2e^{\alpha t}) = \frac{\log(2)}{\alpha} + t.$$

  Moreover, since $\log(1 + t) \leq t$ for $t > 0$, it holds $\log(1 + e^{\alpha t}) \leq e^{\alpha t}$ for every $t \in \mathbb{R}$. Thus it holds that $0 \leq \sigma_\alpha(t) \leq \frac{e^{\alpha t}}{\alpha}$ for every $t < 0$. Therefore $\sigma_\alpha$ satisfies the second condition of Assumption 3.2 for $\rho_1 = 1/\alpha, \rho_2 = \alpha, \rho_3 = 1, \rho_4 = \log(2)/\alpha$.

### A.2    PROOF OF THEOREM 3.4

To prove Theorem 3.4, we first show that Assumption 3.1 can be transported from the input to the output layer.

**Lemma A.1** *Let Assumption 3.1 hold for the training sample. Consider a standard deep CNN architecture which satisfies the following*

1. *The first layer is either convolutional or fully connected while all the other layers can be convolutional, fully connected or max-pooling*

2. *$(\sigma_1, \ldots, \sigma_L)$ are continuous and non-constant activation functions*

*Then for every layer $1 \leq k \leq L$, there exist a set of parameters of the first $k$ layers $(W_l, b_l)_{l=1}^k$ such that it holds $f_k^p(x_i) \neq f_k^q(x_j)$ for every $p, q \in [P_k], i, j \in [N], i \neq j$. Moreover, $(W_l, b_l)_{l=1}^k$ can be chosen in such a way that, except for max-pooling layers, all the weight matrices $U_l = \mathcal{M}_l(W_l)$ have full rank for every $1 \leq l \leq k$.*

**Proof:**    Since Assumption 3.1 holds for the training inputs, one can first pick $(W_1, b_1)$ such that Assumption 3.1 holds at the first layer. Then given that Assumption 3.1 is satisfied at the first layer, the selection of the parameters of the higher layers can be done similarly.

Since our definition of a convolutional layer includes fully connected layer as a special case, it is sufficient to prove the result for the general convolutional structure. Since the first layer is a convolutional layer by our assumption, we denote by $Q = [a^1, \ldots, a^{T_1}] \in \mathbb{R}^{l_0 \times T_1}$ a matrix that

contains the set of convolutional filters of the first layer. Note here that there are $T_1$ filters, namely $\{a^1, \ldots, a^{T_1}\}$, where each filter $a^t \in \mathbb{R}^{l_0}$ for every $t \in [T_1]$. Let us define the set

$$S := \left\{ Q \in \mathbb{R}^{l_0 \times T_1} \mid \mathcal{M}_1(Q) \text{ has low rank} \right\} \cup \bigcup_{\substack{i \neq j \\ p,q \in [P_0] \\ t,t' \in [T_1]}} \left\{ Q \in \mathbb{R}^{l_0 \times T_1} \mid \left\langle a^t, x_i^p \right\rangle - \left\langle a^{t'}, x_j^q \right\rangle = 0 \right\}.$$

Basically, $S$ is the set of "true" parameter matrices of the first layer where the corresponding weight matrix $\mathcal{M}_1(Q)$ has low rank or there exists two patches of two different training samples that have the same inner product with some corresponding two filters. By Assumption 3.1 it holds that $x_i^p \neq x_j^q$ for every $p, q \in [P_0], i \neq j$, and thus the right hand side in the above formula of $S$ is just the union of a finite number of hyperplanes which has Lebesgue measure zero. For the left hand side, it follows from Lemma 2.5 that the set of $Q$ for which $\mathcal{M}_1(Q)$ does not have full rank has measure zero. Thus the left hand side of $S$ is a set of measure zero. Since $S$ is the union of two measure zero sets, it has also measure zero, and thus the complementary set $\mathbb{R}^{l_0 \times T_1} \setminus S$ must be non-empty and we choose $W_1 \in \mathbb{R}^{l_0 \times T_1} \setminus S$.

Since $\sigma_1$ is a continuous and non-constant function, there exists an interval $(\mu_1, \mu_2)$ such that $\sigma_1$ is bijective on $(\mu_1, \mu_2)$. We select and fix some matrix $Q = [a^1, \ldots, a^{T_1}] \in \mathbb{R}^{l_0 \times T_1} \setminus S$ and select some $\beta \in (\mu_1, \mu_2)$. Let $\alpha > 0$ be a free variable and $W_1 = [w_1^1, \ldots, w_1^{T_1}]$ where $w_1^t$ denotes the $t$-th filter of the first layer. Let us pick

$$w_1^t = \alpha Q_{:t} = \alpha a^t, \quad (b_1)_h = \beta, \quad \forall t \in [T_1], h \in [n_1].$$

It follows that $W_1 = \alpha Q$ and thus $\mathcal{M}_1(W_1) = \mathcal{M}_1(\alpha Q) = \alpha \mathcal{M}_1(Q)$ as $\mathcal{M}_1$ is a linear map by our definition. Since $Q \notin S$ by construction, it holds that $\mathcal{M}_1(W_1)$ has full rank for every $\alpha \neq 0$. By Definition 2.1, it holds for every $i \in [N], p \in [P_0], t \in [T_1], h = (p-1)T_1 + t$ that

$$f_1(x_i)_h = \sigma_1(\langle w_1^t, x_i^p \rangle + (b_1)_h) = \sigma_1(\alpha \langle a^t, x_i^p \rangle + \beta).$$

Since $\beta \in (\mu_1, \mu_2)$, one can choose a sufficiently small positive $\alpha$ such that it holds $\alpha \langle a, x_i^p \rangle + \beta \in (\mu_1, \mu_2)$ for every $i \in [N], p \in [P_0], t \in [T_1]$. Under this construction, we will show that every entry of $f_1(x_i)$ must be different from every entry of $f_1(x_j)$ for $i \neq j$. Indeed, let us compare $f_1(x_i)_h$ and $f_1(x_j)_v$ for some $h = (p-1)T_1 + t, v = (q-1)T_1 + t'$ and $i \neq j$. It holds for sufficient small $\alpha > 0$ that

$$f_1(x_i)_h - f_1(x_j)_v = \sigma_1\left(\alpha \left\langle a^t, x_i^p \right\rangle + \beta\right) - \sigma_1\left(\alpha \left\langle a^{t'}, x_j^q \right\rangle + \beta\right) \neq 0 \qquad (8)$$

where the last inequality follows from three facts. First, it holds $\langle a^t, x_i^p \rangle \neq \left\langle a^{t'}, x_j^q \right\rangle$ since $Q \notin S$.

Second, for the chosen $\alpha$ the values of the arguments of the activation function $\sigma_1$ lie within $(\mu_1, \mu_2)$. Third, since $\sigma_1$ is bijective on $(\mu_1, \mu_2)$, it maps different inputs to different outputs.

Now, since the entries of $f_1(x_i)$ and that of $f_1(x_j)$ are already pairwise different from each other, their corresponding patches must be also different from each other no matter how the patches are organized in the architecture, that is,

$$f_1^p(x_i) \neq f_1^q(x_j) \quad \forall p, q \in [P_1], i, j \in [N], i \neq j.$$

Now, if the network has only one layer, *i.e.* $L = 1$, then we are done. Otherwise, we will prove via induction that this property can be translated to any higher layer. In particular, suppose that one has already constructed $(W_l, b_l)_{l=1}^k$ so that it holds

$$f_k(x_i)_h - f_k(x_j)_v \neq 0 \quad \forall h, v \in [n_k], i, j \in [N], i \neq j. \qquad (9)$$

This is true for $k = 1$ due to (8). We will show below that (9) can also hold at layer $k + 1$.

1. Case 1: Layer $k + 1$ is convolutional or fully connected.

   Since (9) holds for $k$ by our induction assumption, it must hold that

   $$f_k^p(x_i) \neq f_k^q(x_j) \quad \forall p, q \in [P_k], i, j \in [N], i \neq j.$$

   which means Assumption 3.1 also holds for the set of features at layer $k$. Thus one can follows the similar construction as done for layer 1 above by considering the output of layer $k$ as input to layer $k + 1$. Then one obtains that there exist $(W_{k+1}, b_{k+1})$ where $U_{k+1} = \mathcal{M}_{k+1}(W_{k+1})$ has full rank so that the similar inequality (8) now holds for layer $k + 1$, which thus implies (9) holds for $k + 1$.

2. Case 2: layer $k + 1$ is max-pooling

By Definition 2.3, it holds $n_{k+1} = P_k$ and one has for every $p \in [P_k]$

$$f_{k+1}(x)_p = \max \left( (f_k^p(x))_1, \ldots, (f_k^p(x))_{l_k} \right).$$

Since (9) holds at layer $k$ by our induction assumption, every entry of every patch of $f_k(x_i)$ must be different from every entry of every patch of $f_k(x_j)$ for every $i \neq j$, that is, $(f_k^p(x_i))_r \neq (f_k^q(x_j))_s$ for every $r, s \in [l_k], p, q \in [P_k], i \neq j$. Therefore, their maximum elements cannot be the same, that is,

$$f_{k+1}(x_i)_p \neq f_{k+1}(x_j)_q \quad \forall p, q \in [n_{k+1}], i, j \in [N], i \neq j.$$

which proves (9) for layer $k + 1$.

So far, we have proved that (9) holds for every $1 \leq k \leq L$. Thus it follows that for every layer $k$, there exists a set of parameters of the first $k$ layers for which the patches at layer $k$ of different training samples are pairwise different from each other, that is, $f_k^p(x_i) \neq f_k^q(x_j)$ for every $p, q \in [P_k], i \neq j$. Moreover, except for max-pooling layers, all the weight matrices up to layer $k$ have been chosen to have full rank. $\square$

**Proof of Theorem 3.4**  Let $A = F_k = [f_k(x_1)^T, \ldots, f_k(x_N)^T] \in \mathbb{R}^{N \times n_k}$. Since our definition of a convolutional layer includes fully connected layer as a special case, it is sufficient to prove the result for convolutional structure. By Theorem's assumption, layer $k$ is convolutional and thus it holds by Definition 2.1 that

$$A_{ij} = f_k(x_i)_j = \sigma \left( \left\langle w_k^t, f_{k-1}^p(x_i) \right\rangle + (b_k)_j \right)$$

for every $i \in [N], t \in [T_k], p \in [P_{k-1}]$ and $j = (p, t) := (p - 1)T_k + t \in [n_k]$.

In the following, we show that there exists a set of parameters of the network such that $rank(A) = N$ and all the weight matrices $U_l = \mathcal{M}_l(W_l)$ have full rank.

First, one observes that the subnetwork consisting of all the layers from the input layer till layer $k - 1$ satisfies the conditions of Lemma A.1. Thus by applying Lemma A.1 to this subnetwork, one obtains that there exist $(W_l, b_l)_{l=1}^{k-1}$ for which all the matrices $(U_l)_{l=1}^{k-1}$, except for max-pooling layers, have full rank and it holds that $f_{k-1}^p(x_i) \neq f_{k-1}^q(x_j)$ for every $p, q \in [P_{k-1}], i \neq j$. The main idea now is to fix the parameters of these layers and pick $(W_k, b_k)$ such that $U_k = \mathcal{M}_k(W_k)$ has full rank and it holds $rank(A) = N$. Let us define the set

$$S = \bigcup_{i \neq j} \bigcup_{p \in [P_{k-1}]} \left\{ a \in \mathbb{R}^{l_{k-1}} \mid \left\langle a, f_{k-1}^p(x_i) - f_{k-1}^p(x_j) \right\rangle = 0 \right\}.$$

From the above construction, it holds that $f_{k-1}^p(x_i) \neq f_{k-1}^p(x_j)$ for every $p \in [P_{k-1}], i \neq j$, and thus $S$ is the union of a finite number of hyperplanes which thus has measure zero. Let us denote by $Q = [a^1, \ldots, a^{T_k}] \in \mathbb{R}^{l_{k-1} \times T_k}$ a parameter matrix that contains all the convolutional filters of layer $k$ in its columns. Pick $a^t \in \mathbb{R}^{l_{k-1}} \setminus S$ for every $t \in [T_k]$, so that it holds that $U_k = \mathcal{M}_k(Q)$ has full rank. Note here that such matrix $Q$ always exists. Indeed, $Q$ is chosen from a positive measure set as its columns (*i.e.* $a^t$) are picked from a positive measure set. Moreover, the set of matrices $Q$ for which $\mathcal{M}_k(Q)$ has low rank has just measure zero due to Lemma 2.5. Thus there always exists at least one matrix $Q$ so that all of its columns do not belong to $S$ and that $\mathcal{M}_k(Q)$ has full rank. In the rest of the proof, the value of matrix $Q$ is fixed. Let $\alpha \in \mathbb{R}$ be a free parameter. Since $\sigma_k$ is a continuous and non-constant function, there exist a $\beta \in \mathbb{R}$ such that $\sigma_k(\beta) \neq 0$. Let the value of $\beta$ be fixed as well. We construct the convolutional filters $W_k = [w_k^1, \ldots, w_k^{T_k}]$ and the biases $b_k \in \mathbb{R}^{n_k}$ of layer $k$ as follows. For every $p \in [P_{k-1}], t \in [T_k], j = (p, t)$, we define

$$w_k^t = -\alpha a^t,$$
$$(b_k)_j = \alpha \left\langle a^t, f_{k-1}^p(x_j) \right\rangle + \beta.$$

It follows that $W_k = -\alpha Q$ and thus $U_k = \mathcal{M}_k(W_k) = -\alpha \mathcal{M}_k(Q)$ as $\mathcal{M}_k$ is a linear map. Moreover, since $\mathcal{M}_k(Q)$ has full rank by construction, it holds that $U_k$ has full rank for every

$\alpha \neq 0$. As $\alpha$ varies, we get a family of matrices $A(\alpha) \in \mathbb{R}^{N \times n_k}$ where it holds for every $i \in [N], j = (p, t) \in [n_k]$ that

$$A(\alpha)_{ij} = \sigma_k \Big( \langle w_k^t, f_{k-1}^p(x_i) \rangle + (b_k)_j \Big) = \sigma_k \Big( \alpha \langle a^t, f_{k-1}^p(x_j) - f_{k-1}^p(x_i) \rangle + \beta \Big). \qquad (10)$$

Note that each row of $A(\alpha)$ corresponds to one training sample and that permutations of the rows of $A(\alpha)$ do not change the rank of $A(\alpha)$. We construct a permutation $\gamma$ of $\{1, 2, \ldots, N\}$ as follows. For every $j = 1, 2, \ldots, N$, let $(p, t)$ be the tuple determined by $j$ (the inverse transformation for given $j \in [n_k]$ is $p = \left\lceil \frac{j}{T_k} \right\rceil$ and $t = j - \left( \left\lceil \frac{j}{T_k} \right\rceil - 1 \right) T_k$) and define

$$\gamma(j) = \underset{i \in \{1,2,\ldots,N\} \setminus \{\gamma(1),\ldots,\gamma(j-1)\}}{\arg \min} \langle a^t, f_{k-1}^p(x_i) \rangle.$$

Note that $\gamma(j)$ is unique for every $1 \leq j \leq N$ since $a^t \notin S$. It is clear that $\gamma$ constructed as above is a permutation of $\{1, 2, \ldots, N\}$ since every time a different element is taken from the index set $[N]$. From the definition of $\gamma(j)$, it holds that for every $j = (p, t) \in [N], i \in [N], i > j$ that

$$\langle a^t, f_{k-1}^p(x_{\gamma(j)}) \rangle < \langle a^t, f_{k-1}^p(x_{\gamma(i)}) \rangle.$$

We can relabel the training data according to the permutation so that w.l.o.g we can assume that $\gamma$ is the identity permutation, that is, $\gamma(j) = j$ for every $j \in [N]$, in which case it holds for every $j = (p, t) \in [N], i \in [N], i > j$ that

$$\langle a^t, f_{k-1}^p(x_j) \rangle < \langle a^t, f_{k-1}^p(x_i) \rangle. \qquad (11)$$

Under the above construction of $(W_l, b_l)_{l=1}^k$, we are ready to show that there exist $\alpha$ for which $rank(A(\alpha)) = N$. Since $\sigma_k$ satisfies Assumption 3.2, we consider the following cases.

1. Case 1: There are finite constants $\mu_+, \mu_- \in \mathbb{R}$ s.t. $\lim_{t \to -\infty} \sigma_k(t) = \mu_-$ and $\lim_{t \to \infty} \sigma_k(t) = \mu_+$ and $\mu_+ \mu_- = 0$.

   Let us consider the first case where $\mu_- = 0$. From (10) and (11) one obtains

   $$\lim_{\alpha \to +\infty} A(\alpha)_{ij} = \begin{cases} \sigma_k(\beta) & j = i \\ \mu_- = 0 & i > j \\ \eta_{ij} & i < j \end{cases} \qquad (12)$$

   where $\eta_{ij}$ is given for every $i < j$ where $j = (p, t)$ as

   $$\eta_{ij} = \begin{cases} \mu_-, & \langle a^t, f_{k-1}^p(x_j) - f_{k-1}^p(x_i) \rangle < 0 \\ \mu_+, & \langle a^t, f_{k-1}^p(x_j) - f_{k-1}^p(x_i) \rangle > 0 \end{cases}$$

   Note that $\eta_{ij}$ cannot be zero for $i \neq j$ because $a^t \notin S$. In the following, let us denote $A(\alpha)_{1:N,1:N}$ as a sub-matrix of $A(\alpha)$ that consists of the first $N$ rows and columns. By the Leibniz-formula one has

   $$\det(A(\alpha)_{1:N,1:N}) = \sigma_k(\beta)^N + \sum_{\pi \in S_N \setminus \{\gamma\}} \text{sign}(\pi) \prod_{j=1}^N A(\alpha)_{\pi(j)j} \qquad (13)$$

   where $S_N$ is the set of all $N!$ permutations of the set $\{1, \ldots, N\}$ and $\gamma$ is the identity permutation. Now, one observes that for every permutation $\pi \neq \gamma$, there always exists at least one component $j$ where $\pi(j) > j$ in which case it follows from (12) that

   $$\lim_{\alpha \to \infty} \prod_{j=1}^N A(\alpha)_{\pi(j)j} = 0.$$

   Since there are only finitely many such terms in (13), one obtains

   $$\lim_{\alpha \to \infty} \det(A(\alpha)_{1:N,1:N}) = \sigma_k(\beta)^N \neq 0$$

   where the last inequality follows from our choice of $\beta$. Since $\det(A(\alpha)_{1:N,1:N})$ is a continuous function of $\alpha$, there exists $\alpha_0 \in \mathbb{R}$ such that for every $\alpha \geq \alpha_0$ it

holds $\det(A(\alpha)_{1:N,1:N}) \neq 0$ and thus $rank(A(\alpha)_{1:N,1:N}) = N$ which further implies $rank(A(\alpha)) = N$. Thus the corresponding set of feature vectors $\{f_k(x_1), \ldots, f_k(x_N)\}$ are linearly independent.

For the case where $\mu_+ = 0$, one can argue similarly. The only difference is that one considers now the limit for $\alpha \to -\infty$. In particular, (10) and (11) lead to

$$\lim_{\alpha \to -\infty} A(\alpha)_{ij} = \begin{cases} \sigma_k(\beta) & i = j \\ \mu_+ = 0 & i > j \\ \eta_{ij} = 0 & i < j. \end{cases}$$

For every permutation $\pi \neq \gamma$ there always exists at least one component $j$ where $\pi(j) > j$, in which case it holds that

$$\lim_{\alpha \to -\infty} \prod_{j=1}^{N} A(\alpha)_{\pi(j)j} = 0.$$

and thus it follows from the Leibniz formula that

$$\lim_{\alpha \to -\infty} \det(A(\alpha)_{1:N,1:N}) = \sigma_k(\beta)^N \neq 0.$$

Since $\det(A(\alpha)_{1:N,1:N})$ is a continuous function of $\alpha$, there exists $\alpha_0 \in \mathbb{R}$ such that for every $\alpha \leq \alpha_0$ it holds $\det(A(\alpha)_{1:N,1:N}) \neq 0$ and thus $rank(A(\alpha)_{1:N,1:N}) = N$ which further implies $rank(A(\alpha)) = N$. Thus the set of feature vectors at layer $k$ are linearly independent.

2. Case 2: There are positive constants $\rho_1, \rho_2, \rho_3, \rho_4$ s.t. $|\sigma_k(t)| \leq \rho_1 e^{\rho_2 t}$ for $t < 0$ and $|\sigma_k(t)| \leq \rho_3 t + \rho_4$ for $t \geq 0$.
Our proof strategy is essentially similar to the previous case. Indeed, for every permutation $\pi \neq \gamma$ there always exist at least one component $j = (p, t) \in [N]$ where $\pi(j) > j$ in which case $\delta_j := \langle a^t, f_{k-1}^p(x_j) - f_{k-1}^p(x_{\pi(j)}) \rangle < 0$ due to (11). For sufficiently large $\alpha > 0$, it holds that $\alpha \delta_j + \beta < 0$ and thus one obtains from (10) that

$$|A(\alpha)_{\pi(j)j}| = |\sigma_k(\alpha \delta_j + \beta)| \leq \rho_1 e^{\rho_2 \beta} e^{-\alpha \rho_2 |\delta_j|}.$$

If $\pi(j) = j$ then $|A(\alpha)_{\pi(j)j}| = |\sigma_k(\beta)|$ which is a constant. For $\pi(j) < j$, one notices that $\delta_j := \langle a^t, f_{k-1}^p(x_j) - f_{k-1}^p(x_{\pi(j)}) \rangle$ can only be either positive or negative as $a^t \notin S$. In this case, if $\delta_j < 0$ then it can be bounded by the similar exponential term as above for sufficiently large $\alpha$. Otherwise it holds $\alpha \delta_j + \beta > 0$ for sufficiently large $\alpha > 0$ and we get

$$|A(\alpha)_{\pi(j)j}| = |\sigma(\alpha \delta_j + \beta)| \leq \rho_3 \delta_j \alpha + \rho_3 \beta + \rho_4.$$

Overall, for sufficiently large $\alpha > 0$, there must exist positive constants $P, Q, R, S, T$ such that it holds for every $\pi \in S_N \setminus \{\gamma\}$ that

$$\left| \prod_{j=1}^{N} A(\alpha)_{\pi(j)j} \right| \leq R(P\alpha + Q)^S e^{-\alpha T}.$$

The upper bound goes to zero as $\alpha$ goes to $\infty$. This combined with the Leibniz formula from (13), we get $\lim_{\alpha \to \infty} \det(A(\alpha)_{1:N,1:N}) = \sigma_k(\beta)^N \neq 0$. Since $\det(A(\alpha)_{1:N,1:N})$ is a continuous function of $\alpha$, there exists $\alpha_0 \in \mathbb{R}$ such that for every $\alpha \geq \alpha_0$ it holds $\det(A(\alpha)_{1:N,1:N}) \neq 0$ and thus $rank(A(\alpha)_{1:N,1:N}) = N$ which implies $rank(A(\alpha)) = N$. Thus the set of feature vectors at layer $k$ are linearly independent.

Overall, we have shown that there always exist $(W_l, b_l)_{l=1}^k$ such that the set of feature vectors $\{f_k(x_1), \ldots, f_k(x_N)\}$ at layer $k$ are linearly independent. Moreover, $(W_l, b_l)_{l=1}^k$ have been chosen so that all the weight matrices $U_l = \mathcal{M}_l(W_l)$, except for max-pooling layers, have full rank for every $1 \leq l \leq k$.

## A.3 PROOF OF THEOREM 3.5

To prove Theorem 3.5, the following key property of real analytic functions is required.

**Lemma A.2** *Nguyen (2015); Mityagin (2015) If $f : \mathbb{R}^n \to \mathbb{R}$ is a real analytic function which is not identically zero then the set $\{x \in \mathbb{R}^n \mid f(x) = 0\}$ has Lebesgue measure zero.*

Any linear function is real analytic and the set of real analytic functions is closed under addition, multiplication and composition, see e.g. Prop. 2.2.2 and Prop. 2.2.8 in Krantz & Parks (2002). As we assume that all the activation functions of the first $k$ layers are real analytic, we get that the function $f_k$ is a real analytic function of $(W_l, b_l)_{l=1}^k$ as it is the composition of real analytic functions. Now, we recall from our definition that $F_k = [f_k(x_1), \ldots, f_k(x_N)]^T \in \mathbb{R}^{N \times n_k}$ is the output matrix at layer $k$ for all training samples. One observes that the set of low rank matrices $F_k$ can be characterized by a system of equations such that all the $\binom{n_k}{N}$ determinants of all $N \times N$ sub-matrices of $F_k$ are zero. As the determinant is a polynomial in the entries of the matrix and thus an analytic function of the entries and composition of analytic functions are again analytic, we conclude that each determinant is an analytic function of the network parameters of the first $k$ layers. By Theorem 3.4, there exists at least one set of parameters of the first $k$ layers such that one of these determinant functions is not identically zero and thus by Lemma A.2, the set of network parameters where this determinant is zero has Lebesgue measure zero. But as all submatrices need to have low rank in order that $rank(F_k) < N$, it follows that the set of parameters where $rank(F_k) < N$ has Lebesgue measure zero.

# B MISSING PROOFS OF SECTION 4

## B.1 PROOF OF LEMMA 4.3

To prove Lemma 4.3, we first derive standard backpropagation in Lemma B.1. In the following we use the Hadamard product $\circ$, which for $A, B \in \mathbb{R}^{m \times n}$ is defined as $A \circ B \in \mathbb{R}^{m \times n}$ with $(A \circ B)_{ij} = A_{ij} B_{ij}$. Let $\delta_{kj}(x_i) = \frac{\partial \Phi}{\partial g_{kj}(x_i)}$ be the derivative of $\Phi$ w.r.t. the value of unit $j$ at layer $k$ evaluated at a single sample $x_i$. We arrange these vectors for all training samples into a single matrix $\Delta_k = [\delta_{k:}(x_1), \ldots, \delta_{k:}(x_N)]^T \in \mathbb{R}^{N \times n_k}$. The following lemma is a slight modification of Lemma 2.1 in (Nguyen & Hein, 2017) for which we provide the proof for completeness.

**Lemma B.1** *Given some hidden layer $1 \le k \le L - 1$. Let $(\sigma_{k+1}, \ldots, \sigma_{L-1})$ be differentiable. Then the following hold:*

1. $\Delta_l = \begin{cases} F_L - Y, & l = L \\ (\Delta_{l+1} U_{l+1}^T) \circ \sigma_l'(G_l), & k + 1 \le l \le L - 1 \end{cases}$

2. $\nabla_{U_l} \Phi = F_{l-1}^T \Delta_l, \ \forall \, k + 1 \le l \le L$

**Proof:**

1. By definition, it holds for every $i \in [N], j \in [n_L]$ that

$$(\Delta_L)_{ij} = \delta_{Lj}(x_i) = \frac{\partial \Phi}{\partial g_{Lj}(x_i)} = f_{Lj}(x_i) - y_{ij}$$

   and hence, $\Delta_L = F_L - Y$.
   For every $k + 1 \le l \le L - 1$, the output of the network for a single training sample can be written as the composition of differentiable functions (*i.e.* the outputs of all layers from $l + 1$ till the output layer), and thus the chain rule yields for every $i \in [N], j \in [n_l]$ that

$$
\begin{aligned}
(\Delta_l)_{ij} = \delta_{lj}(x_i) = \frac{\partial \Phi}{\partial g_{lj}(x_i)} &= \sum_{s=1}^{n_{l+1}} \frac{\partial \Phi}{\partial g_{(l+1)s}(x_i)} \frac{\partial g_{(l+1)s}(x_i)}{\partial f_{lj}(x_i)} \frac{\partial f_{lj}(x_i)}{\partial g_{lj}(x_i)} \\
&= \sum_{s=1}^{n_{l+1}} \delta_{(l+1)s}(x_i)(U_{l+1})_{js} \sigma'(g_{lj}(x_i)) \\
&= \sum_{s=1}^{n_{l+1}} (\Delta_{(l+1)})_{is} (U_{l+1})_{sj}^T \sigma'((G_l)_{ij})
\end{aligned}
$$

   and hence $\Delta_l = (\Delta_{l+1} U_{l+1}^T) \circ \sigma'(G_l)$.

2. For every $k + 1 \leq l \leq L - 1, r \in [n_{l-1}], s \in [n_l]$, one has

$$\frac{\partial \Phi}{\partial (U_l)_{rs}} = \sum_{i=1}^{N} \frac{\partial \Phi}{\partial g_{ls}(x_i)} \frac{\partial g_{ls}(x_i)}{\partial (U_l)_{rs}} = \sum_{i=1}^{N} \delta_{ls}(x_i) f_{(l-1)r}(x_i) = \sum_{i=1}^{N} (F_{l-1}^T)_{ri}(\Delta_l)_{is}$$

$$= \left(F_{l-1}^T \Delta_l\right)_{rs}$$

and hence $\nabla_{U_l} \Phi = F_{l-1}^T \Delta_l$.

$\square$

The following straightforward inequalities are also helpful to prove Lemma 4.3. Let $\lambda_{\min}(\cdot)$ and $\lambda_{\max}(\cdot)$ denotes the smallest and largest eigenvalue of a matrix.

**Lemma B.2** *Let $A \in \mathbb{R}^{m \times n}$ with $m \geq n$. Then it holds $\sigma_{max}(A) \|x\|_2 \geq \|Ax\|_2 \geq \sigma_{min}(A) \|x\|_2$ for every $x \in \mathbb{R}^n$.*

**Proof:** Since $m \geq n$, it holds that $\sigma_{\min}(A) = \sqrt{\lambda_{\min}(A^T A)} = \sqrt{\min \frac{x^T A^T A x}{x^T x}} = \min \frac{\|Ax\|_2}{\|x\|_2}$ and thus $\sigma_{\min}(A) \leq \frac{\|Ax\|_2}{\|x\|_2}$ for every $x \in \mathbb{R}^n$. Similarly, it holds $\sigma_{\max}(A) = \sqrt{\lambda_{\max}(A^T A)} = \sqrt{\max \frac{x^T A^T A x}{x^T x}} = \max \frac{\|Ax\|_2}{\|x\|_2}$ and thus $\sigma_{\max}(A) \geq \frac{\|Ax\|_2}{\|x\|_2}$ for every $x \in \mathbb{R}^n$. $\square$

**Lemma B.3** *Let $A \in \mathbb{R}^{m \times n}, B \in \mathbb{R}^{n \times p}$ with $m \geq n$. Then it holds $\sigma_{max}(A) \|B\|_F \geq \|AB\|_F \geq \sigma_{min}(A) \|B\|_F$.*

**Proof:** Since $m \geq n$, it holds that $\lambda_{\min}(A^T A) = \sigma_{\min}(A)^2$ and $\lambda_{\max}(A^T A) = \sigma_{\max}(A)^2$. Thus we have $\|AB\|_F^2 = \text{tr}(B^T A^T A B) \geq \lambda_{\min}(A^T A) \text{tr}(B^T B) = \sigma_{\min}(A)^2 \|B\|_F^2$. Similarly, it holds $\|AB\|_F^2 = \text{tr}(B^T A^T A B) \leq \lambda_{\max}(A^T A) \text{tr}(B^T B) = \sigma_{\max}(A)^2 \|B\|_F^2$. $\square$

**Proof of Lemma 4.3** We first prove the lower bound. Let $\mathbb{I}_m$ denotes an $m$-by-$m$ identity matrix and $\otimes$ the Kronecker product. From Lemma B.1 it holds $\nabla_{U_{k+1}} \Phi = F_L^T \Delta_{k+1}$ and thus

$$vec(\nabla_{U_{k+1}} \Phi) = (\mathbb{I}_{n_{k+1}} \otimes F_k^T) \, vec(\Delta_{k+1}).$$

It follows that

$$\left\|\nabla_{U_{k+1}} \Phi\right\|_F = \left\|(\mathbb{I}_{n_{k+1}} \otimes F_k^T) \, vec(\Delta_{k+1})\right\|_2 \geq \sigma_{\min}(F_k) \|vec(\Delta_{k+1})\|_2 = \sigma_{\min}(F_k) \|\Delta_{k+1}\|_F$$

$$(14)$$

where the inequality follows from Lemma B.2 for the matrix $(\mathbb{I}_{n_{k+1}} \otimes F_k^T) \in \mathbb{R}^{n_k n_{k+1} \times N n_{k+1}}$ with $n_k \geq N$ by Assumption 4.1. Using Lemma B.1 again, one has

$$\|\Delta_{k+1}\|_F = \left\|(\Delta_{k+2} U_{k+2}^T) \circ \sigma'_{k+1}(G_{k+1})\right\|_F$$

$$\geq \left\|\sigma'_{k+1}(G_{k+1})\right\|_{\min} \left\|\Delta_{k+2} U_{k+2}^T\right\|_F$$

$$= \left\|\sigma'_{k+1}(G_{k+1})\right\|_{\min} \left\|U_{k+2} \Delta_{k+2}^T\right\|_F$$

$$\geq \left\|\sigma'_{k+1}(G_{k+1})\right\|_{\min} \sigma_{\min}(U_{k+2}) \|\Delta_{k+2}\|_F$$

where the last inequality follows from Lemma B.3 for the matrices $U_{k+2} \in \mathbb{R}^{n_{k+1} \times n_{k+2}}$ and $\Delta_{k+2}^T$ with $n_{k+1} \geq n_{k+2}$ by Assumption 4.1. By repeating this for $\|\Delta_{k+2}\|_F, \ldots, \|\Delta_{L-1}\|_F$, one gets

$$\|\Delta_{k+1}\|_F \geq \Big( \prod_{l=k+1}^{L-1} \|\sigma'_l(G_l)\|_{\min} \sigma_{\min}(U_{l+1}) \Big) \|\Delta_L\|_F = \Big( \prod_{l=k+1}^{L-1} \|\sigma'_l(G_l)\|_{\min} \sigma_{\min}(U_{l+1}) \Big) \|F_L - Y\|_F$$

$$(15)$$

From (14), (15), one obtains

$$\left\|\nabla_{U_{k+1}} \Phi\right\|_F \geq \sigma_{\min}(F_k) \Big( \prod_{l=k+1}^{L-1} \|\sigma'_l(G_l)\|_{\min} \sigma_{\min}(U_{l+1}) \Big) \|F_L - Y\|_F$$

which proves the lower bound.

The proof for upper bound is similar. Indeed one has

$$\left\|\nabla_{U_{k+1}}\Phi\right\|_F = \left\|(\mathbb{I}_{n_{k+1}} \otimes F_k^T) \, vec(\Delta_{k+1})\right\|_2 \leq \sigma_{\max}(F_k) \left\|vec(\Delta_{k+1})\right\|_2 = \sigma_{\max}(F_k) \left\|\Delta_{k+1}\right\|_F \tag{16}$$

where the inequality follows from Lemma B.2 Now, one has

$$\begin{aligned}
\left\|\Delta_{k+1}\right\|_F &= \left\|(\Delta_{k+2}U_{k+2}^T) \circ \sigma_{k+1}'(G_{k+1})\right\|_F \\
&\leq \left\|\sigma_{k+1}'(G_{k+1})\right\|_{\max} \left\|\Delta_{k+2}U_{k+2}^T\right\|_F \\
&= \left\|\sigma_{k+1}'(G_{k+1})\right\|_{\max} \left\|U_{k+2}\Delta_{k+2}^T\right\|_F \\
&\leq \left\|\sigma_{k+1}'(G_{k+1})\right\|_{\max} \sigma_{\max}(U_{k+2}) \left\|\Delta_{k+2}\right\|_F
\end{aligned}$$

where the last inequality follows from Lemma B.3. By repeating this chain of inequalities for $\left\|\Delta_{k+2}\right\|_F, \ldots, \left\|\Delta_{L-1}\right\|_F$, one obtains:

$$\left\|\Delta_{k+1}\right\|_F \leq \Big( \prod_{l=k+1}^{L-1} \left\|\sigma_l'(G_l)\right\|_{\max} \sigma_{\max}(U_{l+1}) \Big) \left\|\Delta_L\right\|_F = \Big( \prod_{l=k+1}^{L-1} \left\|\sigma_l'(G_l)\right\|_{\max} \sigma_{\max}(U_{l+1}) \Big) \left\|F_L - Y\right\|_F . \tag{17}$$

From (16), (17), one obtains that

$$\left\|\nabla_{U_{k+1}}\Phi\right\|_F \leq \sigma_{\max}(F_k) \Big( \prod_{l=k+1}^{L-1} \left\|\sigma_l'(G_l)\right\|_{\max} \sigma_{\max}(U_{l+1}) \Big) \left\|F_L - Y\right\|_F$$

which proves the upper bound.

## B.2 PROOF OF THEOREM 4.5

1. Since layer $k + 1$ is fully connected, it holds at every critical point in $S_k$ that $\nabla_{W_{k+1}}\Phi = 0 = \nabla_{U_{k+1}}\Phi$. This combined with Theorem 4.4 yields the result.

2. One basically needs to show that there exist $(W_l, b_l)_{l=1}^L$ such that it holds: $\Phi\Big((W_l, b_l)_{l=1}^L\Big) = 0, rank(F_k) = N$ and $U_l = \mathcal{M}_l(W_l)$ has full rank $\forall l \in [k+2, L]$

   Note that the last two conditions are fulfilled by the fact that $(W_l, b_l)_{l=1}^L \in S_k$.

   By Assumption 4.1, the subnetwork consisting of the first $k$ layers satisfies the condition of Theorem 3.4. Thus by applying Theorem 3.4 to this subnetwork, one obtains that there exist $(W_l, b_l)_{l=1}^k$ such that $rank(F_k) = N$. In the following, we fix these layers and show how to pick $(W_l, b_l)_{l=k+1}^L$ such that $F_L = Y$. The main idea now is to make the output of all the training samples of the same class become identical at layer $k + 1$ and thus they will have the same network output. Since there are only $m$ classes, there would be only $m$ distinct outputs for all the training samples at layer $L - 1$. Thus if one can make these $m$ distinct outputs become linearly independent at layer $L-1$ then there always exists a weight matrix $W_L$ that realizes the target output $Y$ as the last layer is just a linear map by assumption. Moreover, we will show that, except for max-pooling layers, all the parameters of other layers in the network can be chosen in such a way that all the weight matrices $(U_l)_{l=k+2}^L$ achieve full rank. Our proof details are as follows.

   Case 1: $k = L - 1$
   It holds $rank(F_{L-1}) = N$. Pick $b_L = 0$ and $W_L = F_{L-1}^T(F_{L-1}F_{L-1}^T)^{-1}Y$. Since the output layer is fully connected, it follows from Definition 2.2 that $F_L = F_{L-1}W_L + \mathbf{1}_N b_L^T = Y$. Since $rank(F_k) = N$ and the full rank condition on $(W_l)_{l=k+2}^L$ is not active when $k = L - 1$, it holds that $(W_l, b_l)_{l=1}^L \in S_k$ which finishes the proof.

   Case 2: $k = L - 2$
   It holds $rank(F_{L-2}) = N$. Let $A \in \mathbb{R}^{m \times n_{L-1}}$ be a full row rank matrix such that $A_{ij} \in range(\sigma_{L-1})$. Note that $n_{L-1} \geq n_L = m$ due to Assumption 4.1. Let $D \in \mathbb{R}^{N \times n_{L-1}}$

be a matrix satisfying $D_{i:} = A_{j:}$ whenever $x_i$ belongs to class $j$ for every $i \in [N], j \in [m]$. By construction, $F_{L-2}$ has full row rank, thus we can pick $b_{L-1} = 0, W_{L-1} = F_{L-2}^T(F_{L-2}F_{L-2}^T)^{-1}\sigma_{L-1}^{-1}(D)$. Since layer $L - 1$ is fully connected by assumption, it follows from Definition 2.2 that $F_{L-1} = \sigma_{L-1}(F_{L-2}W_{L-1} + \mathbf{1}_N b_{L-1}^T) = D$ and thus $(F_{L-1})_{i:} = D_{i:} = A_{j:}$ whenever $x_i$ belongs to class $j$.

So far, our construction of the first $L - 1$ layers has led to the fact that all the training samples belonging to the same class will have identical output at layer $L - 1$. Since $A$ has full row rank by construction, we can pick for the last layer $b_L = 0, W_L = A^T(AA^T)^{-1}Z$ where $Z \in \mathbb{R}^{m \times m}$ is our class embedding matrix with $rank(Z) = m$. One can easily check that $AW_L = Z$ and that $F_L = F_{L-1}W_L + \mathbf{1}_N b_L^T = F_{L-1}W_L$ where the later follows from Definition 2.2 as the output layer is fully connected. Now one can verify that $F_L = Y$. Indeed, whenever $x_i$ belongs to class $j$ one has

$$(F_L)_{i:} = (F_{L-1})_{i:}^T W_L = (A)_{j:}^T W_L = Z_{j:} = Y_{i:}.$$

Moreover, since $rank(F_k) = N$ and $rank(W_L) = rank(A^T(AA^T)^{-1}Z) = m$, it holds that $(W_l, b_l)_{l=1}^L \in S_k$. Therefore, there exists $(W_l, b_l)_{l=1}^L \in S_k$ with $\Phi\left((W_l, b_l)_{l=1}^L\right) = 0$.

Case 3: $k \leq L - 3$
It holds $rank(F_k) = N$. Let $E \in \mathbb{R}^{m \times n_{k+1}}$ be any matrix such that $E_{ij} \in \text{range}(\sigma_{k+1})$ and $E_{ip} \neq E_{jq}$ for every $1 \leq i \neq j \leq N, 1 \leq p, q \leq n_{k+1}$. Let $D \in \mathbb{R}^{N \times n_{k+1}}$ satisfies $D_{i:} = E_{j:}$ for every $x_i$ from class $j$. Pick $b_{k+1} = 0, W_{k+1} = F_k^T(F_kF_k^T)^{-1}\sigma_{k+1}^{-1}(D)$. Note that the matrix is invertible as $F_k$ has been chosen to have full row rank. Since layer $k + 1$ is fully connected by our assumption, it follows from Definition 2.2 that $F_{k+1} = \sigma_{k+1}(F_kW_{k+1} + \mathbf{1}_N b_{k+1}^T) = \sigma_{k+1}(F_kW_{k+1}) = D$ and thus it holds

$$(F_{k+1})_{i:} = D_{i:} = E_{j:} \tag{18}$$

for every $x_i$ from class $j$.

So far, our construction has led to the fact that all training samples belonging to the same class have identical output at layer $k + 1$. The idea now is to see $E$ as a new training data matrix of a subnetwork consisting of all layers from $k + 1$ till the output layer $L$. In particular, layer $k + 1$ can be seen as the input layer of this subnetwork and similarly, layer $L$ can be seen as the output layer. Moreover, every row of $E \in \mathbb{R}^{m \times n_{k+1}}$ can be seen as a new training sample to this subnetwork. One can see that this subnet together with the training data matrix $E$ satisfy the conditions of Theorem 3.4 at the last hidden layer $L - 1$. In particular, it holds that

- The rows of $E \in \mathbb{R}^{m \times n_{k+1}}$ are componentwise different from each other, and thus the input patches must be also different from each other, and thus $E$ satisfies Assumption 3.1

- Every layer from $k+1$ till $L-1$ is convolutional or fully connected due to Assumption 4.1

- The width of layer $L - 1$ is larger than the number of samples due to Assumption 4.1, that is, $n_{L-1} \geq n_L = m$

- $(\sigma_{k+2}, \ldots, \sigma_{L-1})$ satisfy Assumption 3.2 due to Assumption 4.1

By applying Theorem 3.4 to this subnetwork and training data $E$, we obtain that there must exist $(W_l, b_l)_{l=k+2}^{L-1}$ for which all the weight matrices $(U_l)_{l=k+2}^{L-1}$ have full rank such that the set of corresponding $m$ outputs at layer $L - 1$ are linearly independent. In particular, let $A \in \mathbb{R}^{m \times n_{L-1}}$ be the corresponding outputs of $E$ through this subnetwork then it holds that $rank(A) = m$. Intuitively, if one feeds $E_{j:}$ as an input at layer $k + 1$ then one would get $A_{j:}$ as an output at layer $L - 1$. This combined with (18) leads to the fact that if one now feeds $(F_{k+1})_{i:} = E_{j:}$ as an input at layer $k + 1$ then one would get at layer $L - 1$ the output $(F_{L-1})_{i:} = A_{j:}$ whenever $x_i$ belongs to class $j$.

Last, we pick $b_L = 0, W_L = A^T(AA^T)^{-1}Z$. It follows that $AW_L = Z$. Since the output layer $L$ is fully connected, it holds from Definition 2.2 that $F_L = F_{L-1}W_L + \mathbf{1}_N b_L^T = F_{L-1}W_L$.

One can verify now that $F_L = Y$. Indeed, for every sample $x_i$ from class $j$ it holds that

$$(F_L)_{i:} = (F_{L-1})_{i:}^T W_L = A_{j:}^T W_L = Z_{j:} = Y_{i:}.$$

Overall, we have shown that $\Phi = 0$. In addition, it holds $rank(F_k) = N$ from the construction of the first $k$ layers. All the matrices $(U_l)_{l=k+2}^{L-1}$ have full rank by the construction of the subnetwork from $k+1$ till $L$. Moreover, $U_L = W_L$ also has full rank since $rank(W_L) = rank(A^T (AA^T)^{-1} Z) = m$. Therefore it holds $(W_l, b_l)_{l=1}^L \in S_k$.

## B.3 PROOF OF COROLLARY 4.6

It is clear that the network structure satisfies the conditions of Theorem 4.4 as every layer is fully connected. Moreover, the input patches also satisfy Assumption 3.1 because every input patch is simply one training sample in this case but since there are no identical training samples, one derives that the input patches must be different from each other.

Since all the conditions of Theorem 4.4 are met, the application of Theorem 4.4 yields the result.

