# OpenReview forum: "The loss surface and expressivity of deep convolutional neural networks"
_ICLR.cc/2018/Conference — Invite to Workshop Track_

### Official Review · AnonReviewer3 · 2017-11-27
**Review of "The loss surface and expressivity of deep convolutional neural networks"**

**Rating:** 4
**Confidence:** 4

**Review:**

This paper presents several theoretical results on the loss functions of CNNs and fully-connected neural networks. I summarize the results as follows:

(1) Under certain assumptions, if the network contains a "wide“ hidden layer, such that the layer width is larger than the number of training examples, then (with random weights) this layer almost surely extracts linearly independent features for the training examples.

(2) If the wide layer is at the top of all hidden layers, then the neural network can perfectly fit the training data.

(3) Under similar assumptions and within a restricted parameter set S_k, all critical points are the global minimum. These solutions achieve zero squared-loss.

I would consider result (1) as the main result of this paper, because (2) is a direct consequence of (1). Intuitively, (1) is an easy result. Under the assumptions of Theorem 3.5, it is clear that any tiny random perturbation on the weights will make the output linearly independent. The result will be more interesting if the authors can show that the smallest eigenvalue of the output matrix is relatively large, or at least not exponentially small.

Result (3) has severe limitations, because: (a) there can be infinitely many critical point not in S_k that are spurious local minima; (b) Even though these spurious local minima have zero Lebesgue measure, the union of their basins of attraction can have substantial Lebesgue measure; (c) inside S_k, Theorem 4.4 doesn't exclude the solutions with exponentially small gradients, but whose loss function values are bounded away above zero. If an optimization algorithm falls onto these solutions, it will be hard to escape.

Overall, the paper presents several incremental improvement over existing theories. However, the novelty and the technical contribution are not sufficient for securing an acceptance.

---

> ### Author Response · Authors · 2017-12-07
> **Author's replies**
>
> We do not agree with the assessment of novelty and contribution of reviewer 3. Up to our knowledge only the paper of Cohen and Shashua (ICML 2016) analyzes general CNN architectures. As CNN architectures are obviously very important in practice, we think that a better theoretical understanding is urgently needed. Our paper contains two main results. First we show that CNNs used in practice produce linearly independent features (for ImageNet with VGG or Inception architecture) with probability 1 (Theorem 3.5) at the wide layer (first layer in VGG and Inception). We think that this is a very helpful result to understand how and why current CNNs work also with respect to the recent debate around generalization properties of state of the art networks (Zhang et al, 2017). Second, we give necessary and sufficient conditions for global optima under squared loss (Theorem 4.4) and show that all critical points in S_k are globally optimal under the conditions of Theorem 4.5. We think that this is a significant contribution to the theoretical understanding of CNN architectures. In particular, we would like to emphasize that all our results are applicable to the real problem of interest without any simplifying assumptions.
>
> We agree in general with the reviewer that it might be nice to have even stronger results e.g. convergence of gradient descent/SGD to the global optimum. But given that the current state of the art in this regard is limited to one hidden layer together with additional distributional assumptions and does not cover deep CNNs used in practice (multiple filters, overlapping patches, deep architecture), we think that the reviewer demands too much. Even papers which consider just deep linear models have been appreciated in the community and get very good reviews at ICLR 2018.
>
> Specific answers:
> "Intuitively, (1) is an easy result. Under the assumptions of Theorem 3.5, it is clear that any tiny random perturbation on the weights will make the output linearly independent."
>
> There are a lot of mathematical results which are intuitive but that does not mean that they are easy to prove.
>
> "The result will be more interesting if the authors can show that the smallest eigenvalue of the output matrix is relatively large, or at least not exponentially small."
>
> We agree that this result would be interesting, but one has to start somewhere (see general comment above).
>
> "Result (3) has severe limitations, because: (a) there can be infinitely many critical point not in S_k that are spurious local minima; (b) Even though these spurious local minima have zero Lebesgue measure, the union of their basins of attraction can have substantial Lebesgue measure; (c) inside S_k, Theorem 4.4 doesn't exclude the solutions with exponentially small gradients, but whose loss function values are bounded away above zero. If an optimization algorithm falls onto these solutions, it will be hard to escape."
>
> (a) Yes, but then these critical points not in S_k (the complement of S_k has measure zero) must have either low rank weight matrices in the layers above the wide layer or the features are not linearly independent at the wide layer. We don't see any reason in the properties of the loss which would enforce low rank in the weight matrices of a CNN. Moreover, it seems unlikely that a critical point with a low rank matrix is a suboptimal local minimum as this would imply that all possible full rank perturbations have larger/equal objective (we don't care if the complement of S_k potentially contains additional global minima). Even for simpler models like two layer linear networks, it has been shown by (Baldi and Hornik, 1989) that all the critical points with low rank weight matrices have to be saddle points and thus cannot be suboptimal local minima. See also other parallel submissions at ICLR 2018 for similar results and indications for deep linear models (e.g. Theorem 2.1, 2.2 in https://openreview.net/pdf?id=BJk7Gf-CZ, and Theorem 5 in https://openreview.net/pdf?id=ByxLBMZCb).
> Moreover, a similar argument applies to the case where one has critical point such that the features are not linearly independent at the wide layer. As any neighborhood of such a critical point contains points which have linearly independent features at the wide layer (and thus it is easy to achieve zero loss), it is again unlikely that this critical point is a suboptimal local minimum.
> In summary, if there are any critical points in the complement of S_k, then it is very unlikely that these are suboptimal local minima but they are rather also global minima, saddle points or local maxima.
>
> (b/c) We agree that these are certainly interesting questions but the same comment applies as above. Moreover, we see no reason why critical points with low rank weight matrices should be attractors.

---

### Official Review · AnonReviewer1 · 2017-11-27

**Rating:** 7
**Confidence:** 2

**Review:**

This paper analyzes the expressiveness and loss surface of deep CNN. I think the paper is clearly written, and has some interesting insights.

---

> ### Author Response · Authors · 2017-12-07
> **Author's replies**
>
> Thanks a lot for your reviews. We are happy to answer any additional questions you might have regarding our work.

---

### Official Review · AnonReviewer2 · 2017-12-12
**The loss surface and expressivity of deep convolutional neural networks**

**Rating:** 5
**Confidence:** 2

**Review:**

This paper analyzes the loss function and properties of CNNs with one "wide" layer, i.e., a layer with number of neurons greater than the train sample size. Under this and some additional technique conditions, the paper shows that this layer can extract linearly independent features and all critical points are local minimums. I like the presentation and writing of this paper. However, I find it uneasy to fully evaluate the merit of this paper, mainly because the "wide"-layer assumption seems somewhat artificial and makes the corresponding results somewhat expected. The mathematical intuition is that the severe overfitting induced by the wide layer essentially lifts the loss surface to be extremely flat so training to zero/small error becomes easy. This is not surprising. It would be interesting to make the results more quantitive, e.g., to quantify the tradeoff between having local minimums and having nonzero training error.

---

> ### Author Response · Authors · 2017-12-21
> **Author's replies**
>
> "I like the presentation and writing of this paper. However, I find it uneasy to fully evaluate the merit of this paper, mainly because the "wide"-layer assumption seems somewhat artificial and makes the corresponding results somewhat expected."
>
> Please note Table 1, where we have listed several state-of-the-art CNN networks, which have such a wide layer (more hidden units than the number of training points) in the case of ImageNet. These are VGG, Inception V3 and Inception V4. Thus we don't see why this wide layer assumption is "artificial" if CNNs which had large practical success fulfill this condition.
>
> "The mathematical intuition is that the severe overfitting induced by the wide layer essentially lifts the loss surface to be extremely flat so training to zero/small error becomes easy. This is not surprising."
>
> We think that our finding that practical CNNs such as VGG/Inception produce linearly independent features at the wide layer for ImageNet for almost any weight configuration up to the wide layer is an interesting finding which fosters the understanding of these CNNs. While the fact that whether the result is surprising or not is rather a matter of personal taste, what we find more relevant and important is if this result can help to advance the theoretical understanding of practical networks using rigorous math, which it does.
>
> "It would be interesting to make the results more quantitive, e.g., to quantify the tradeoff between having local minimums and having nonzero training error."
>
> Such results are currently only available for coarse approximations of neural networks where it is not clear how and if they apply to neural networks used in practice. Meanwhile, our results hold exactly for the architectures used in practice.

---

### Official Review · AnonReviewer4 · 2017-12-13
**Interesting direction.**

**Rating:** 6
**Confidence:** 3

**Review:**

This paper presents an analysis of convolutional neural networks from the perspective of how the rank of the features is affected by the kinds of layers found in the most popular networks. Their analysis leads to the formulation of a certain theorem about the global minima with respect to parameters in the latter portion of the network.

The authors ask important questions, but I am not sure that they obtain important answers. On the plus side, I'm glad that people are trying to further our understanding our neural networks, and I think that their investigation is worthy of being published.

They present a collection of assumptions, lemmas, and theorems. They have no choice but to have assumptions, because they want to abstract away the "data" part of the analysis while still being able to use certain properties about the rank of the features at certain layers.

Most of my doubts about this paper come from the feeling that equivalent results could be obtained with a more elegant argument about perturbation theory, instead of something like the proof of Lemma A1. That being said, it's easy to voice such concerns, and I'm willing to believe that there might not exist a simple way to derive the same results with an approach more along the line of "whatever your data, pick whatever small epsilon, and you can always have the desired properties by perturbing your data by that small epsilon in a random direction". Have the authors tried this ?

I'm not sure if the authors were the first to present this approach of analyzing the effects of convolutions from a "patch perspective", but I think this is a clever approach. It simplifies the statement of some of their results. I also like the idea of factoring the argument along the concept of some critical "wide layer".

Good review of the literature.

I wished the paper was easier to read. Some of the concepts could have been illustrated to give the reader some way to visualize the intuitive notions. For example, maybe it would have been interesting to plot the rank of features a every layer for LeNet+MNIST ?

At the end of the day, if a friend asked me to summarize the paper, I would tell them :

"Features are basically full rank. Then they use a square loss and end up with an over-parametrized system, so they can achieve loss zero (i.e. global minimum) with a multitude of parameters values."


Nitpicking :

"This paper is one of the first ones, which studies CNNs."
This sentence is strange to read, but I can understand what the authors mean.

"This is true even if the bottom layers (from input to the wide layer) and chosen randomly with probability one."
There's a certain meaning to "with probability one" when it comes to measure theory. The authors are using it correctly in the rest of the paper, but in this sentence I think they simply mean that something holds if "all" the bottom layers have random features.

---

> ### Author Response · Authors · 2018-01-05
> **Author's replies**
>
> We thank reviewer 4 for the detailed comments.
>
> "They present a collection of assumptions, lemmas, and theorems. They have no choice but to have assumptions, because they want to abstract away the "data" part of the analysis while still being able to use certain properties about the rank of the features at certain layers."
>
> Yes, the reviewer is right, we did not want to make assumptions on the distribution of the training data
> as these assumptions are very difficult to check. Instead our assumptions can all be easily checked for a given training set and CNN architecture.
>
> "Most of my doubts about this paper come from the feeling that equivalent results could be obtained with a more elegant argument about perturbation theory, instead of something like the proof of Lemma A1. That being said, it's easy to voice such concerns, and I'm willing to believe that there might not exist a simple way to derive the same results with an approach more along the line of "whatever your data, pick whatever small epsilon, and you can always have the desired properties by perturbing your data by that small epsilon in a random direction". Have the authors tried this ?"
>
> We don't know but we can prove Lemma A1 for any given dataset (fulfilling the stated assumptions). However, we use a perturbation argument to show that our assumptions on the training data are always fulfilled for an arbitrarily small perturbation of the data (similar to what the reviewer suggests).
>
> "I'm not sure if the authors were the first to present this approach of analyzing the effects of convolutions from a "patch perspective", but I think this is a clever approach. It simplifies the statement of some of their results. I also like the idea of factoring the argument along the concept of some critical "wide layer".
>
> Good review of the literature."
>
> Up to the best of our knowledge we have not seen that this patch argument has been used before. It is a very convenient tool to analyze even much more general CNN architectures than the ones currently used.
>
> "I wished the paper was easier to read. Some of the concepts could have been illustrated to give the reader some way to visualize the intuitive notions. For example, maybe it would have been interesting to plot the rank of features a every layer for LeNet+MNIST ?"
>
> We would be very grateful for pointers where we could improve the readability of the paper. We have added a plot for the architecture of Figure 1, where we vary the number of filters T_1 and plot the rank of the feature at the first convolutional layer. As shown by Theorem 3.5 we get full rank for T_1>=89 which implies n_1>=N for the first convolutional layer. In this case the rank of F_1 is 60000 and training error is zero and the loss is minimized almost up to single precision.  We think that this illustrates nicely the result of Theorem 3.5
>
> " "This paper is one of the first ones, which studies CNNs."
> This sentence is strange to read, but I can understand what the authors mean."
>
> We agree: please check the new uploaded version, where we have changed it to:
> This paper is one of the first ones, which theoretically analyzes deep CNNs
>
> ""This is true even if the bottom layers (from input to the wide layer) and chosen randomly with probability one."
> There's a certain meaning to "with probability one" when it comes to measure theory. The authors are using it correctly in the rest of the paper, but in this sentence I think they simply mean that something holds if "all" the bottom layers have random features."
>
> We agree that this can be misunderstood. What we prove is that it holds for almost any weight configuration for the layers from input to the wide layer with respect to the Lebesgue measure (up to a set of measure zero). As in practice the weights are often initialized using e.g. a Gaussian distribution, we wanted to highlight that our result holds with probability 1. In order to clarify this we have added a footnote "are choosen randomly ("with respect to any probability measure which has a density with respect to the Lebesgue measure"). Thus it holds for any probability measure on the weight space which has a density function. We have changed the uploaded manuscript in that way.

---

### Decision · Program_Chairs · 2018-01-29
**ICLR 2018 Conference Acceptance Decision**

**Decision:**

Invite to Workshop Track

**Comment:**

Dear authors,

While I appreciate the result that a convolutional layer can have full rank output, this allowing a dataset to be classified perfectly under mild conditions, the fact that all reviewers expressed concern about the statement is an indication that the presentation sill needs quite a bit of work.

Thus, I recommend it as an ICLR workshop paper.